# Online Packet Scheduling with Deadlines and Learning

**Gianmarco Genalti** [1]  **Achraf Azize** [2]  **Vianney Perchet** [3]

## Abstract

Network routers that enforce Quality-of-Service (QoS) guarantees must decide, at every clock cycle, which expiring packet of information to transmit, even when the value of the packet is *unknown* until it is processed. We frame this problem as the *Online Packet Scheduling with Deadlines* (OPSD) problem under Partial Feedback: packets arrive at every clock cycle, with different *deadlines*, but the weights are only observed after execution. Under a stochastic assumption on the unknown weights, we explore different variants of the OPSD problem with bandit feedback. We establish a connection between our setting and the *sleeping bandits* problem, and set our learning goal to $\alpha$-regret minimization. We provide algorithms with provable $\alpha$-regret guarantees under different spans of slackness, distinguishing systems allowing for randomization and systems that do not. In every scenario, our algorithms achieve an $\alpha$-regret upper bound of $\widetilde{\mathcal{O}}\left(\sqrt{KT}\right)$, matching the lower bound for the standard bandit setting. In the practically relevant case of 2-bounded deadline instances, where the deadline is set at most one clock cycle away from the arrival, our deterministic algorithm achieves the provably tightest possible competitive ratio. Remarkably, when the number of distinct packet types $K \geq 2$ is finite, it is possible to break the well-established $\Phi = \frac{1+\sqrt{5}}{2}$ competitive ratio barrier and attain a tighter competitive ratio $\theta_K$ ranging in $\left[\sqrt{2}, \Phi\right)$.

## 1 Introduction

The *Online Packet Scheduling with Deadlines* (OPSD, for short) problem, also called *bounded delay buffer problem* or *traffic shaping*, is a theoretical framework that has been introduced to model the behavior of buffers that store data packets in *Quality-of-Service* (QoS) networks (Karakus & Durresi, 2017). In QoS networks, it is crucial to enforce service standards, especially on the performance as seen by the end-users. QoS plays a key role in the modern digital infrastructure, notable examples include sensor networks (Iyer & Kleinrock, 2003), cloud computing (Szigeti et al., 2013) and Intelligent Transportation Systems (ITS) (Belamri et al., 2021).

In its simpler version, the OPSD framework assumes that packets with unit processing cost arrive over time and are stored in an unlimited buffer. In OPSD, the presence of *deadlines* models the practical scenario of low-latency systems. In other words, it is not possible to keep a packet idling in the buffer for too many clock cycles. Every packet must then either be processed before its requested deadline or discarded from the buffer. The model is completed by including scores representing the importance of each packet, called *weights*. Of course, any algorithm `ALG` has the goal of maximizing its total score, or *cumulative gain*. Naturally, the presence of the weights makes the problem non-trivial and models the QoS property of the system. The arrival sequence defines an *instance* of the OPSD problem and can be chosen in an arbitrary way by an *adversary*. As different instances may present different opportunities to `ALG`, a standard way to evaluate its performance is to compare it to the best possible decision-maker for that instance—called `OPT`—through the *competitive ratio* $C$, which is the ratio between the gain of `OPT` and the gain of `ALG`.

Over the last few years, there has been a surge in interest in *learning-augmented* online algorithms. The idea is to include *online learning techniques* to deal with reduced information or to boost the performance of online algorithms. Examples of online optimization problems with predictions include OPSD (Liang et al., 2023), facility location (Agrawal et al., 2022), and paging (Bansal et al., 2022). Our work initiates the study of a variant of the OPSD problem, which we call *Stochastic Online K-Packet Scheduling with Deadlines* ($K$-OPSD, for short), where we include online learning (and, in particular, *bandit learning*) into the standard OPSD problem. In the $K$-OPSD framework, we assume that every packet belongs to one of $K$ different types, which is known by the scheduler. However, the weight of a packet is not revealed until it is processed. We make a

[1]Politecnico di Milano [2]FairPlay Joint Team, CREST, ENSAE, IP Paris [3]FairPlay Joint Team, CREST, ENSAE, IP Paris, CRITEO AI Team. Correspondence to: Gianmarco Genalti <gianmarco.genalti@polimi.it>.

*Proceedings of the $43^{rd}$ International Conference on Machine Learning*, Seoul, South Korea. PMLR 306, 2026. Copyright 2026 by the author(s).

stochastic assumption, assuming that every packet type is associated with a probability distribution, and when a packet is processed, the weight is sampled from the corresponding distribution. Recently, this type of extension appeared in (Merlis et al., 2023) in the context of preemptive scheduling and in (Levy et al., 2024) for online paging.

The goal of a scheduler in the $K$-OPSD problem can be naturally extended from the classic OPSD setting: a scheduling algorithm ALG wants to maximize its *expected* cumulative gain. In $K$-OPSD, a scheduling algorithm has less information available than in OPSD. As learning becomes a necessary component for a scheduler, a natural way to evaluate its performance is $\alpha$-regret. The notion of $\alpha$-regret combines the competitive ratio with an additive regret w.r.t. an optimal decision-maker that knows the average weights for every packet type and does not need to learn them.

The stochastic MAB problem (Lattimore & Szepesvári, 2020) shares many similarities with the $K$-OPSD problem. We show that the $K$-OPSD problem includes (and generalizes) the Sleeping Bandit problem (Kleinberg et al., 2010), which is a generalization of the classic MAB problem where an adversary can choose the subset of available actions at every round. The OPSD setting is included in the $K$-OPSD setting, and even in the simpler scenarios, it is possible to prove a lower bound on the competitive ratio of any algorithm. For example, even in 2-*bounded instances* where the deadline of every packet is upper bounded by the arrival time plus one, a lower bound on the competitive ratio is $\Phi = \frac{1+\sqrt{5}}{2} \approx 1.618$, the so-called *golden ratio*. A competitive ratio greater than one implies the impossibility of obtaining a sub-linear *standard* regret in the $K$-OPSD problem. In this paper, we address the following question:

> *Is it possible to devise $\alpha$-no-regret algorithms in some $K$-OPSD problems?*

We answer this question positively. It turns out that in many relevant scenarios this is possible, and with provably tight regret rates and state-of-the-art competitive ratios. Interestingly, the $K$ types assumption can even be leveraged to improve the existing competitive ratios.

**Motivating Example** As previously discussed, the OPSD problem is well-motivated by several applications. We now state a real-world scenario in which $K$-OPSD is a better fit, which motivates the setting. We are in a digital advertising scenario, in which there exist $K$ ad publishers that, in a really fast-paced manner, ask to participate in an auction to display an ad on their space. As an advertiser, you are interested in participating in the auctions with the highest possible probability of receiving a click, which is, however, unknown. However, the requests are often so large in number that not all of them can be processed. If you don't participate in the auction, you will never know if you would

have received a click or not. Moreover, auctions have a limited lifespan and usually finish after a few seconds. This problem combines reward estimation and scheduling with deadlines, since a quick decision based on past observation must be made. Moreover, as the lifespan of auctions is usually *really* short, it is even a stronger motivation for scenarios with short deadlines, such as the 2-bounded $K$-OPSD problem.

## 1.1 Existing Results and Summary of Contributions

In Table 1, we summarize the existing results in the classic OPSD problem and compare them to our results. The $K$-OPSD setting is novel and unexplored in the literature, thus our bounds on the $\alpha$-regret can only be compared in light of the existing competitive ratios and upper bounds on the regret. However, in the 2-bounded setting, we improve over existing results when $K < \infty$. As customary in this literature, we distinguish between *deterministic* and *randomized* algorithms. In what follows, we report an overview of our results. Through the paper, we use the $\mathcal{O}$ notation to hide universal constants, and the $\widetilde{\mathcal{O}}$ notation to hide universal constants and logarithms.

In the first part of this paper, we propose deterministic algorithms for the OPSD and the $K$-OPSD setting.

- In the 2-bounded and 3-bounded $K$-OPSD setting, we propose a deterministic algorithm that suffers a $\Phi$-regret upper bounded as $\widetilde{\mathcal{O}}\left(\sqrt{KT}\right)$ (Theorem 3.1). This upper bound is also provided in an instance-dependent version. In classic Sleeping MABs the best attainable bound on the the regret is in the order of $\widetilde{\mathcal{O}}\left(\sqrt{KT}\right)$ as well, and the competitive ratio matches the best existing one. This result highlights the connection between the $K$-OPSD setting and the Sleeping MAB setting, and the analysis provides insights into how the learning procedure can be decomposed in a meaningful way from the traditional competitive ratio analysis.

- In the 2-bounded OPSD setting, when the number of distinct weights is $K < \infty$, we provide a novel deterministic algorithm called $\text{ALG}^\theta$ that achieves a competitive ratio of $\theta_K$ (4.1) smaller than the best known competitive ratio of $\Phi$ (Hajek, 2001). Note that $\theta_K \in \left[\sqrt{2}, \Phi\right)$ for $K < \infty$ and $\theta_\infty = \Phi$. This upper bound on the competitive ratio is tight according to the lower bound proposed in (Hajek, 2001).

- In the 2-bounded $K$-OPSD setting, we adapt $\text{ALG}^\theta$ to deal with learning, resulting in $\text{ALG}^{\theta,U}$. This deterministic algorithm suffers a $\theta_K$-regret bounded as $\widetilde{\mathcal{O}}\left(\sqrt{KT}\right)$ (Theorem 4.2), and we prove that this is nearly tight with a lower bound (Theorem 4.3).

We dedicate the second part of this paper to randomized

| Setting | Algorithm Type | OPSD Competitive Ratio | | $K$-OPSD $\alpha$-regret |
|---|---|---|---|---|
| 1-bounded (MAB) | Det. (upper) | 1 | | $R_{1,T} \le \widetilde{\mathcal{O}}\left(\sqrt{KT}\right)$ (Kleinberg et al., 2010) |
| | Det. (lower) | 1 | | $R_{1,T} \ge \Omega\left(\sqrt{KT}\right)$ (Kleinberg et al., 2010) |
| 2-bounded | Det. (upper) | $\theta_K \overset{K\to\infty}{\to} \Phi$, ($\Phi$ (Kesselman et al., 2001)) | | $R_{\theta_K,T} \le \widetilde{\mathcal{O}}\left(\sqrt{KT}\right)$, $\quad \theta_K < \Phi$ |
| | Det. (lower) | $\theta_K \overset{K\to\infty}{\to} \Phi$ (Hajek, 2001) | | $R_{\theta_K,T} \ge \Omega\left(\sqrt{T}\right)$, $\quad \theta_{K-1} < \Phi$ |
| | Rand. (upper) | $\frac{5}{4}$ (Chin et al., 2006) | | $R_{\frac{5}{4},T} \le \widetilde{\mathcal{O}}\left(\sqrt{KT}\right)$ |
| | Rand. (lower) | $\frac{5}{4}$ (Chin & Fung, 2003) | | N.A. |
| 3-bounded | Det. (upper) | $\Phi$ (Kesselman et al., 2001) | | $R_{\Phi,T} \le \widetilde{\mathcal{O}}\left(\sqrt{KT}\right)$ |
| | Det. (lower) | $\Phi$ | | $R_{\theta_K,T} \ge \Omega\left(\sqrt{T}\right)$, $\quad \theta_{K-1} < \Phi$ |
| Unbounded Slackness | Det. (upper) | $\Phi$ (Veselỳ et al., 2022) | | N.A. |
| | Det. (lower) | $\Phi$ | | $R_{\theta_K,T} \ge \Omega\left(\sqrt{T}\right)$, $\theta_{K-1} < \Phi$ |
| | Rand. (upper) | $\frac{e}{e-1}$ (Chin et al., 2006) | | $R_{\frac{e}{e-1},T} \le \widetilde{\mathcal{O}}\left(\sqrt{KT}\right)$ |
| | Rand. (lower) | $\frac{5}{4}$ | | N.A. |

*Table 1.* Comparison of existing results with ours. Results without citations are direct consequences of other entries. Results highlighted in gray are our contributions. The $\widetilde{\mathcal{O}}$ notation hides universal constants and logarithmic factors. The $\Omega$ notation hides universal constants.

algorithms.

- In the 2-bounded $K$-OPSD setting, we propose $\mathtt{ALG}^{R2}$, a randomized algorithm that suffers a $\frac{5}{4}$-regret upper bounded as $\widetilde{\mathcal{O}}\left(\sqrt{KT}\right)$ (Theorem 5.1).

- In the $K$-OPSD setting with unbounded slackness, we propose $\mathtt{ALG}^{Rs}$, an algorithm that suffers a $\frac{e}{e-1}$-regret upper bounded as $\widetilde{\mathcal{O}}\left(\sqrt{KT}\right)$ (Theorem 5.2).

In both cases, our bounds match the best existing competitive ratio as well as the tightest regret attainable in Sleeping MABs.

## 2 Setting and Technical Tools

In Online Packet Scheduling with Deadlines (OPSD), an instance is a sequence of packets $\mathcal{P} = \{(r_p, d_p, w_p)\}_p$, where $r_p$ and $d_p \ge r_p$ are integers representing the arrival time and deadline of the packet $p$, respectively. $w_p \ge 0$ is a real number that represents the *weight* of packet $p$. For the rest of this paper, we will slightly abuse notation by calling $w_p$ simply $p$. The time is discrete and divided into slots. A *schedule* assigns a subset of packets $\mathcal{S} \subset \mathcal{P}$ to time slots such that (i) any packet $p \in \mathcal{S}$ is assigned to a slot in $\{r_p, \ldots, d_p\}$ and (ii) each slot has at most one packet assigned. The objective is to design a schedule that maximizes the total weight of the packets in $\mathcal{S}$. In the *s-bounded* variant of the problem, the set of instances is constrained to those where $d_p \le r_p + s - 1$, *i.e.*, every packet can be assigned in either the arrival slot or one of the next $s - 1$ ones (for a total of $s$ possibilities). We call $T \coloneqq \max_{p \in \mathcal{P}} d_p$. An online algorithm $\mathtt{ALG}$ observes

the packets arriving over time and, at every round $t \in [T]$, has to decide which packet to schedule without observing the future. The packets not scheduled at time $t$ are stored in a set $\mathcal{B}_t$ called *buffer*, until their deadline.

**Competitive Ratio** We call $G_{\mathtt{ALG}}$ the total gain of $\mathtt{ALG}$, *i.e.* the total sum of the packets scheduled up to $T$. The standard performance metric in the literature is the *competitive ratio*, *i.e.*, the ratio between the performance of a benchmark decision-maker and $G_{\mathtt{ALG}}$ (Borodin & El-Yaniv, 2005). The goal is to minimize the competitive ratio. The adversary observes the algorithm and decides the input sequence. If the algorithm is deterministic, the full sequence can be chosen beforehand, as the behavior of the algorithm is completely predictable. In the case of randomized algorithms, the adversary cannot observe its random bits during the execution, but can decide the next entry in the sequence based on the past decisions of the algorithm. The benchmark is the best possible decision-maker that observes the whole sequence, called $\mathtt{OPT}$. Thus, the competitive ratio $C$ is defined by the inequality $G_{\mathtt{OPT}} \le C \cdot G_{\mathtt{ALG}}$.

### 2.1 Learning in Online Packet Scheduling with Deadlines

We consider the *Stochastic Bounded-Delay $K$-Packet Scheduling* problem, a variant of the problem where there exist $K$ distinct classes of packets, and all packets belonging to the same class have their weights sampled from the same probability distribution with mean $\mu_k$ for $k \in [K]$. We then define an instance as $\mathcal{P} = \{(r_p, d_p, X_p, c_p)\}_p$, where $c_p \in [K]$ indicates the class of the packet $p$ and $X_p$ is the

weight of packet $p$, which is sampled from the distribution $\mathcal{D}_{c_p}$ with mean $w_p \coloneqq \mu_{c_p}$. Without loss of generality, we order the expectations of the $K$ types in a descending order, *i.e.*, $\mu_1 > \mu_2 > \ldots > \mu_K$. Moreover, we define $\Delta_{i,j} \coloneqq \mu_i - \mu_j$ for $j \geq i$ as the *sub-optimality gap* of $j$ w.r.t. $i$. We will slightly abuse the notation and indicate $w_p$ simply as $p$ for the rest of this paper. The scheduler does not observe the distributions nor their means, and observes $X_p \in [0,1]$ at time $t \in [T]$ only if the packet $p$ is assigned to slot $t$. The adversary is allowed to decide the type of the packet injected, but not the realization of the weight (and does not observe it beforehand). This variant can be trivially extended to the $s$-bounded setting. As it will become handy in what follows, we introduce the set $V_t \subseteq [K]$ as the set of packet types such that at least one packet of the type is in the buffer and has a deadline $t$. We call $B_t \subseteq [K]$ the set of packet types that are not in $V_t$ and such that at least one packet of the type is in the buffer and has a deadline *greater than $t$*. Note that no reasonable ALG schedules a packet of type $i$ with deadline greater than $t$ if $i \in V_t$.

$\alpha$**-Regret** First, consider a deterministic algorithm ALG. Let $G_{\text{ALG}} = \sum_{i \in \mathcal{S}} X_i$ be the total gain of ALG. We assume that no adversary can observe the actual realizations of the weights, and consider an OPT that does the best possible scheduling in expectation, *i.e.* uses the weight's expectation as the actual value. We call $\mathbb{E}[G_{\text{OPT}}]$ the expected total gain of OPT. Thus, a natural performance metric is the $\alpha$-*regret*, defined as $R_{\alpha,T} = \mathbb{E}[G_{\text{OPT}}] - \alpha\mathbb{E}[G_{\text{ALG}}]$, where the expectation is taken w.r.t. the weights' randomization (that is, in this case, the only possible source of randomness). If ALG is randomized, the expected total gain also accounts for the algorithm's randomization. The definition of OPT is the same: OPT can always be assumed deterministic as the provided sequence is fixed. Finally, we say that an algorithm achieves $\alpha$-*no-regret* if $R_{\alpha,T}^{\text{ALG}} = o(T)$, *i.e.* its $\alpha$-regret is sublinear in $T$.

**Deterministic vs Randomized Algorithms in $K$-OPSD**
In $K$-OPSD we need to set a formal definition of a deterministic algorithm. Indeed, due to the stochastic nature of the weights, the algorithm's decision is intrinsically stochastic. Thus, as customary in the bandits literature, we say that an algorithm is deterministic if its decision at time $t$ is *predictable* w.r.t. the filtration up to time $t-1$. In other words, the adversary can predict which packet is processed by ALG at time $t$ by just observing the history until $t-1$. For instance, the well-known UCB1 algorithm (Auer et al., 2002) is deterministic in this sense. One can interpret the distinction between deterministic and randomized algorithms as a distinction between the strength of the adversary: while competitive ratios are lower in randomized algorithms, it is also true that the adversary is less informed. An adversary observing the actual random bits of the algorithm would

result in every algorithm being, *de facto*, deterministic.

**Mean Estimation and Confidence Bounds** Define $\widehat{\mu}_{i,t} = \frac{1}{N_i(t)}\sum_{s=1}^{t} r_i \mathbb{1}_{p_s=i}$ to be the empirical mean of the observed weights from packets of type $i$ observed up to time $t$. Let $\text{LCB}_{i,t}(\delta)$ and $\text{UCB}_{i,t}(\delta)$ be lower and upper confidence bounds on the unknown mean weight from a packet of type $i$, with confidence level $1 - \delta$. Let $\beta_{i,t}(\delta) \coloneqq \frac{1}{2}(\text{UCB}_{i,t}(\delta) - \text{LCB}_{i,t}(\delta))$ be (half of) the width of the confidence bound. We use the same procedure introduced by (Levy et al., 2024) to construct our confidence bounds. Specifically, the confidence width is chosen to be $\beta_{i,t}(\delta) = \sqrt{\frac{\log(KT^2/\delta)}{2N_i(t)}}$, where $K$ is the number of bounds that must hold simultaneously (in our case, one per packet type). Then, $\text{UCB}_{i,t}(\delta) \triangleq \min\{\text{UCB}_{i,t-1}(\delta), \widehat{\mu}_{i,t} + \beta_{i,t}(\delta)\}$, and similarly, $\text{LCB}_{i,t}(\delta) \triangleq \max\{\text{LCB}_{i,t-1}(\delta), \widehat{\mu}_{i,t} - \beta_{i,t}(\delta)\}$. Lemma C.1 in (Levy et al., 2024) ensures that the true weight lies inside the confidence bound with probability at least $1 - \frac{1}{KT}$ for a proper choice of $\delta$, and that the UCBs and LCBs are positive and monotonic. For clarity, we will omit the dependence on $\delta$ (except when necessary) and assume that $\delta \leq \frac{1}{T}$. $\delta$ is an algorithm parameter (it is necessary to compute UCBs and LCBs), and additional details on the choice of $\delta$ for every algorithm can be found in the proofs.

## 2.2 Connection with Multi-Armed Bandits

In the Sleeping bandit problem (Kleinberg et al., 2010), a learner is sequentially faced with an *adversarially* chosen set of actions $\mathcal{A}_t \subset [K]$ for $T$ rounds. Each action is associated with a probability distribution that the learner does not observe, and a sample (or *reward*) is received after the corresponding action is chosen. The cumulative reward of the learner is given by the sum of all rewards obtained up to $T$. This problem, which is a generalization of the stochastic $K$-armed bandit problem, has been explored in previous literature under the lens of *regret minimization*. In regret minimization, the learner's goal is to minimize the performance gap w.r.t. the best possible learner, *i.e.* $R_T \coloneqq \mathbb{E}[G_{\text{OPT}}] - \mathbb{E}[G_{\text{ALG}}]$ (Lattimore & Szepesvári, 2020). Note that regret minimization is a special case of $\alpha$-regret minimization ($\alpha = 1$). As the sleeping bandit problem includes the stochastic $K$-armed bandit problem, the lower bound on the regret that any algorithm can suffer is greater or equal, and it is in the order of $\Omega(\sqrt{KT})$. Interestingly, the $K$-OPSD problem is a generalization of the $K$-*armed* sleeping bandit learning problem.

**Lemma 2.1** (Sleeping Bandits are a special case of $K$-OPSD)**.** *Every 1-bounded instance of the $K$-OPSD problem can be reduced to a $K$-armed Sleeping Bandit problem, and vice versa.*

The AUER algorithm (Kleinberg et al., 2010), a gold standard for regret minimization in sleeping bandits, employs

**Algorithm 1** $\text{EDF}_{\Phi}^{L}$

1: **for** $t \in [T]$ **do**
2:   Receive new packets, identify $h_t = \max_{h \in \mathcal{B}_t} UCB_{h,t}$.
3:   Schedule $f_t \in \arg\min_{f \in \mathcal{B}_t \text{ s.t. } \Phi UCB_{f,t} \geq UCB_{h,t}} \{d_f\}$
4: **end for**

the well-known *optimism in the face of uncertainty* principle (Auer et al., 2002) to achieve no-regret. AUER suffers an expected regret upper bounded by $\widetilde{\mathcal{O}}\left(\sqrt{KT}\right)$.

## 3 Instance-Dependent Guarantees in $2$ and $3$-bounded $K$-OPSD

It is known that a greedy (w.r.t. to deadlines) algorithm can always obtain a competitive ratio of 2, for every $s > 1$ (Hajek, 2001). However, it is possible to improve this value by slightly modifying the greedy algorithm. We recall the $\beta$-Ratio Earliest Deadline First ($\text{EDF}_{\beta}$) algorithm (Kesselman et al., 2001). If $h_t$ is the heaviest packet in $\mathcal{B}_t$, this algorithm schedules the packet $f_t$ with the earliest deadline s.t. $\beta f_t \geq h_t$. This family of algorithms is also called *thresholding* algorithms, since the decision is only taken by scanning the packets in a deadline-ordered fashion until a threshold is exceeded.

We show that a simple adaptation of $\text{EDF}_{\beta}$ achieves $\Phi$-no-regret in 2 and 3-bounded $K$-OPSD. In particular, we show that combining the optimism in the face of the uncertainty principle with the $\text{EDF}_{\Phi}$ algorithm, it is possible to obtain both instance-dependent and worst-case $\Phi$-regret bounds. We call $\text{EDF}_{\Phi}^{L}$ the resulting algorithm, and we report its pseudocode in Algorithm 1. At every round, the algorithm identifies the packet type in the buffer with the largest UCB, *i.e.*, $h_t = \max_{h \in \mathcal{B}_t} UCB_{h,t}$, and then sends the smallest deadline packet $f_t$ s.t.

$$f_t \in \underset{f \in \mathcal{B}_t \text{ s.t. } \Phi UCB_{f,t} \geq UCB_{h,t}}{\arg\min} \{d_f\}.$$

In stochastic MABs, it is possible to provide *instance-dependent* guarantees on the regret that depend on the time horizon $T$ and on the reward-generating distributions, in particular on the sub-optimality gaps. In Sleeping MABs, it is possible to upper bound the regret of AUER as $\mathcal{O}\left(\sum_{i=1}^{K-1} \frac{\ln T}{\Delta_{i,i+1}}\right)$, which is tight (Kleinberg et al., 2010). These types of guarantees are interesting because they provide additional insights on which situations are harder to deal with for an algorithm. In the following proposition, we bound the $\Phi$-regret of $\text{EDF}_{\Phi}^{L}$ in both an instance-dependent and worst-case fashion.

**Proposition 3.1** ($\Phi$-regret of $\text{EDF}_{\Phi}^{L}$). *In $2$-bounded and $3$-bounded $K$-OPSD instances, $\text{EDF}_{\Phi}^{L}$ suffers an expected cu-*

**Algorithm 2** $\text{ALG}^{\theta}$

**Require:** Number of types $K$.
1: Solve the system (1), get $(\theta_K, x_1, \ldots, x_{K-1})$, and set $j \leftarrow 0$
2: **for** $t \in [T]$ **do**
3:   Receive new packets and update $V_t$ and $B_t$
4:   Identify $b_t$ and $v_t$
5:   **if** $v_t < \frac{x_j}{x_{j+1}} b_t$ **then**
6:     Schedule $b_t$
7:     $j \leftarrow 0$
8:   **else**
9:     Schedule $v_t$
10:    $j \leftarrow \begin{cases} j+1, & \text{if } v_t \leq b_t, \\ 0, & \text{otherwise.} \end{cases}$
11:  **end if**
12: **end for**

*mulative $\Phi$-regret bounded as*

$$R_{\Phi,T}^{\text{EDF}_{\Phi}^{L}} \leq \mathcal{O}\left(K \sum_{i=1}^{K-1} \frac{\ln T}{\Delta_{i,i+1}} + \sum_{\substack{i,j \in [K] \\ \Delta_{i,j}^{\Phi} > 0}} \frac{\ln T}{\Delta_{i,j}^{\Phi}}\right),$$

*where $\Delta_{i,j}^{\Phi} := |\mu_i - \Phi\mu_j|$. Alternatively, the $\Phi$-regret can be bounded as*

$$R_{\Phi,T}^{\text{EDF}_{\Phi}^{L}} \leq \widetilde{\mathcal{O}}\left(\sqrt{KT}\right).$$

The first addendum of the instance-dependent bound is in the same order as the instance-dependent bound of AUER. The second addendum is a similar complexity term that features a $\Phi$-scaled version of the sub-optimality gaps between the types. The worst-case bound shows how $\text{EDF}_{\Phi}^{L}$ achieves a $\Phi$-regret which is bounded in the same order as the regret of AUER.

## 4 Improving the $\Phi$ Competitive Ratio in $2$-Bounded $K$-OPSD

In this section, we focus on 2-bounded instances. Our goal is to provide an algorithm that improves the performance of $\text{EDF}_{\Phi}^{L}$ in 2-bounded $K$-OPSD instances. Thus, our focus will be on the competitive ratio. An idea that may naturally arise is to exploit the fact that there are only $K$ finite weight types. In bounded-delay packet scheduling, the lower bound of $\Phi$ for the competitive ratio is proven using an instance with infinitely many different packet weights. In our $K$-OPSD learning setting, having $K = \infty$ is unreasonable, as the regret in bandits scales with $\sqrt{K}$ even in the classic setting without deadlines, making the problem non-learnable.

### 4.1 Breaking the $\Phi$ Barrier in $2$-Bounded OPSD, without learning.

We introduce $\text{ALG}^{\theta}$, an algorithm achieving $\theta_K$-competitive ratio, breaking the $\Phi$ barrier for the competitive ratio when

only $K$ types of packets are available. We start by introducing the following system of equations, which will play a key role in the algorithm and its analysis. Let $x_0 = 1$

$$\begin{cases} x_0 = 1, \quad x_1 = \frac{1}{\theta-1}, \\ x_j = \frac{\theta+1}{\theta-1}(x_{j-1} - x_{j-2}), \quad 2 \leq j \leq K-1, \quad (1) \\ x_{K-1} = (\theta + 1)x_{K-2} \end{cases}$$

Equations (1) have been used in Hajek (2001) to prove the $\Phi$ lower bound for the competitive ratio. Indeed, the author proves that there exists a unique nonnegative solution $(\theta_K, x_1, \ldots, x_{K-1})$, where $\theta_K \to \Phi$ as $K$ goes to infinity. For example, when $K = 2$, we have $\theta_2 = \sqrt{2} \approx 1.41$, and for $K = 3$ we have $\theta_3 = \frac{3}{2} = 1.5$. Then the value progressively increases approaching $\Phi \approx 1.618$.

In Algorithm 2 we report the pseudocode of $\texttt{ALG}^\theta$. As a first step, given $K$, $\texttt{ALG}^\theta$ solves the system (1) before the trial, then it operates in epochs. $\texttt{ALG}^\theta$ keeps track of the number of rounds $j$ passed since the current epoch started.

At time $t$, $\texttt{ALG}^\theta$ receives new packets and identifies $v_t \in V_t$ and $b_t \in B_t$ as the heaviest packets in the algorithm's buffer with deadlines $t$ and $t + 1$, respectively. The scheduling rule is the following: if $v_t < \frac{x_j}{x_{j+1}}b_t$ then schedule $b_t$, else schedule $v_t$. The epoch can terminate (setting $j = 0$) in two ways: either $b_t$ is scheduled, or $v_t$ is scheduled and $v_t \geq b_t$.

Note that $\texttt{ALG}^\theta$ needs to know $K$, but not the actual values of the $K$ weights involved in advance. Also, $v_{t+j}$ is always guaranteed to exist in $\texttt{ALG}^\theta$ buffer due to the epoch termination condition. The following proposition upper bounds the competitive ratio of $\texttt{ALG}^\theta$.

**Proposition 4.1** ($\theta_K$- Competitive Ratio for $\texttt{ALG}^\theta$). *In 2-bounded $K$-OPSD instances, $\texttt{ALG}^\theta$ satisfies*

$$G_{OPT} \leq \theta_K G_{ALG^\theta}.$$

Proposition 4.1 shows that, in the case of at most $K$ distinct types of packets, improving over the well-established $\Phi$ competitive ratio is possible. This result is of independent interest, as the scenario with $K$ types has never been explored in the literature of online packet scheduling with bounded deadlines. Also, the lower bound construction from (Hajek, 2001) implies the tightness of this result.

### 4.2 $\theta_K$-regret Upper Bound for 2-Bounded Delay $K$-OPSD, with learning.

The algorithmic idea underlying $\texttt{ALG}^\theta$ can be simply extended to the learning scenario, *i.e.* by using UCBs instead of the actual weights. We now define $v_t$ and $b_t$ according to their UCBs, *i.e.* $v_t \in \arg\max_{v \in V_t} UCB_{v,t}$ and $b_t \in \arg\max_{b \in B_t} UCB_{b,t}$. We call $\texttt{ALG}^{\theta,U}$ the resulting algorithm, which operates as $\texttt{ALG}^\theta$ using weights' estimates. In Algorithm 3, we report the pseudocode of $\texttt{ALG}^{\theta,U}$.

---

**Algorithm 3** $\texttt{ALG}^{\theta,U}$

**Require:** Number of types $K$.
1: Solve the system (1), get $(\theta_K, x_1, \ldots, x_{K-1})$, and initialize $x_0 \leftarrow 1$, $x_K \leftarrow x_{K-1}$ and $j \leftarrow 0$.
2: **for** $t \in [T]$ **do**
3:    **if** j = 0 **then**
4:       Compute UCBs for every packet type, obtaining $\{UCB_{i,t}\}_{i \in [K]}$
5:    **end if**
6:    Receive new packets and update $V_t$ and $B_t$
7:    Identify $v_t \in \arg\max_{v \in V_t}\{UCB_{v,t}\}$
8:    Identify $b_t \in \arg\max_{b \in B_t}\{UCB_{b,t}\}$
9:    **if** $UCB_{v_t,t} < \frac{x_j}{x_{j+1}}UCB_{b_t,t}$ **then**
10:      Schedule $b_t$
11:      $j \leftarrow 0$
12:    **else**
13:      Schedule $v_t$
14:      $j \leftarrow \begin{cases} j+1, & \text{if } UCB_{v_t,t} \leq UCB_{b_t,t}, \\ 0, & \text{otherwise} \end{cases}$
15:    **end if**
16: **end for**

---

With the following result, we prove that $\texttt{ALG}^{\theta,U}$ achieves a tight $\theta_K$-regret bound w.r.t. to the lower bound presented in Theorem 4.3.

**Theorem 4.2** (Upper Bound on the $\theta_K$-regret of $\texttt{ALG}^{\theta,U}$). *For every instance of the stochastic $K$-types packet scheduling problem, we have*

$$\mathbb{E}[G_{OPT}] \leq \theta_K \mathbb{E}[G_{ALG^{\theta,U}}] + \widetilde{\mathcal{O}}\left(\sqrt{KT}\right).$$

*Proof Idea.* The proof builds on the proof of Proposition 4.1. In fact, it can be conducted in almost the same way noticing the the gain of $\texttt{OPT}$ inside an epoch can be upper bounded by the UCB of the gain of $\texttt{ALG}$ inside the same. Then, it is only needed to upper bound the sum of the confidence bounds in the same way as the proof of Proposition 3.1. The full proof can be found in Appendix B.

This result closes the problem of learning in 2-bounded packet scheduling problems with deterministic algorithms, providing a matching $\theta_K$-regret upper bound to the lower bound of Theorem 4.3. Note that this result strictly improves the one in Proposition 3.1.

### 4.3 Regret Lower Bound.

We provide a lower bound on the $\theta_K$-regret for the stochastic $K$-OPSD problem.

**Theorem 4.3** (Lower Bound for $\theta_K$-regret). *In 2-bounded $(K + 1)$-OPSD instances, every $\texttt{ALG}$ must satisfy*

$$\mathbb{E}[G_{OPT}] - \theta_K \mathbb{E}[G_{ALG}] \geq \Omega\left(\sqrt{T}\right).$$

Theorem 4.3 implies that $\texttt{ALG}^\theta$ is worst-case nearly-optimal in this scenario. This result is of particular interest because there are few examples of lower bounds for $\alpha$-regret.

The technical challenge mainly relies on deviating from the lower bound construction for the competitive ratio: the instance used by (Hajek, 2001) is built in a way that, no matter what the algorithm does, the competitive ratio is always exactly $\Phi$. The construction can be adapted, via an early stopping, to get a $\theta_K$ lower bound, but this competitive ratio is always tightly achieved, no matter what the algorithm does. Any modification to the construction yields a situation in which there exists a simple thresholding rule for which the competitive ratio ends up being higher than $\theta_K$. We solved this problem by a *signed-perturbation*, that forces any algorithm, even with adversarially chosen thresholding logics, to $\theta_K$-alpha regret.

## 5 Randomized Algorithms for the $K$-OPSD Problem

In this section, we focus on *randomized* algorithms. Randomized algorithms have been studied less than their deterministic counterparts in the packet scheduling literature. However, these algorithms can achieve better worst-case theoretical guarantees in systems allowing for randomization. Note that the performance of a randomized algorithm cannot really be compared to that of the deterministic algorithm, because randomization makes the adversary *weaker*, as it removes the possibility of perfectly predicting the algorithm's next move. Indeed, even if an algorithm takes random decisions, an adversary that knows its random bits could, in principle, behave as if the algorithm is deterministic.

In the OPSD literature, no algorithm matches the $\frac{5}{4}$ competitive ratio lower bound for randomized algorithms. In the 2-bounded case, the algorithm from (Chin et al., 2006) achieves a tight $\frac{5}{4}$. When $s$ is arbitrary, the known lower bound for the competitive ratio in $s$-bounded instances is still $\frac{5}{4}$ (Chin & Fung, 2003), and the best known algorithm can achieve $\frac{e}{e-1}$ for every $s$ (Chin et al., 2006).

### 5.1 Achieving $\frac{5}{4}$-no-regret in 2-bounded $K$-OPSD.

We start by providing a randomized algorithm with $\frac{5}{4}$-no-regret guarantees for 2-bounded $K$-OPSD. We use the same notation as in the previous sections. For every time $t \in [T]$ and couple of types $a, b \in [K]$, consider the following:

$$\widehat{p}_{ab,t} = \begin{cases} \max\left\{\dfrac{4UCB_{a,t}}{5LCB_{b,t}}, \dfrac{1}{5}\right\}, & UCB_{a,t} \le LCB_{b,t} \\ \dfrac{4}{5}, & UCB_{a,t} > LCB_{b,t} \text{ and } LCB_{a,t} \le UCB_{b,t} \\ 1, & LCB_{a,t} > UCB_{b,t} \end{cases}.$$

Our algorithm, namely $\texttt{ALG}^{R2}$, whose pseudocode is reported in Algorithm 4, works as follows: at round $t$, it selects $a_t \in \arg\max_{a \in V_t} UCB_{a,t}$ and $b_t \in \arg\max_{b \in B_t} UCB_{b,t}$. Then, send a packet of type $a_t$ with probability $\widehat{p}_{a_t b_t, t}$, otherwise send a packet of type $b_t$ (with probability $\widehat{q}_{a_t b_t, t} =$

---

**Algorithm 4** $\texttt{ALG}^{R2}$

1: **for** $t \in [T]$ **do**
2:     Compute UCBs and LCBs for every packet type, obtaining $\{UCB_{i,t}\}_{i \in [K]}$ and $\{LCB_{i,t}\}_{i \in [K]}$
3:     Receive new packets and update $B_t$ and $V_t$
4:     Identify $a_t \in \arg\max_{a \in V_t} UCB_{a,t}$
5:     Identify $b_t \in \arg\max_{b \in B_t} UCB_{b,t}$
6:     **if** $\exists i \in \{a_t, b_t\}$ s.t. $N_i \le 50 \ln T$ **then**
7:         Schedule $i_t \in \arg\min_{i \in \{a_t, b_t\}}\{N_{i,t}\}$
8:     **else**
9:         Compute $\widehat{p}_{a_t b_t, t}$
10:         Schedule $a_t$ with probability $\widehat{p}_{a_t b_t, t}$, else schedule $b_t$
11:     **end if**
12: **end for**

---

$1 - \widehat{p}_{a_t b_t, t}$). The algorithm also performs a *burn-in period*, in which for the first $50 \log T$ times a packet is selected as $a_t$ or $b_t$, it is automatically scheduled.

Finally, we state the main result of this section.

**Theorem 5.1** (Upper Bound for the $\frac{5}{4}$-regret of $\texttt{ALG}^{R2}$). *In 2-bounded $K$-OPSD instances where randomization is allowed, $\texttt{ALG}^{R2}$ satisfies*

$$\mathbb{E}[G_{OPT}] \le \frac{5}{4}\mathbb{E}[G_{ALG^{R2}}] + \widetilde{\mathcal{O}}\left(\sqrt{KT}\right). \quad (2)$$

This result shows that randomization can be effectively leveraged in the 2-bounded $K$-OPSD problem to attain $\alpha$-no-regret guarantees with the best known competitive ratio of $\frac{5}{4}$. It remains an open question whether the $K$ types assumption can be exploited to obtain a further sharpened bound with a competitive ratio smaller than $\frac{5}{4}$ (as we did for deterministic algorithms in Section 4).

### 5.2 Achieving $\frac{e}{e-1}$-regret in $s$-bounded $K$-OPSD.

We now extend our results on randomized algorithms by providing a randomized algorithm with $\frac{e}{e-1}$-no-regret guarantees for the $s$-bounded $K$-OPSD problem (note, $\frac{e}{e-1} \approx 1.582 < \Phi$), for any arbitrary $s \in \mathbb{N}$. We follow the notation introduced in the previous sections. Let $\underline{h}_t \in \arg\max_{h \in \mathcal{B}_t} LCB_{h,t}$ be the packet having the largest LCB in the buffer and $\bar{h}_t \in \arg\max_{h \in \mathcal{B}_t} UCB_{h,t}$ be the packet having the largest UCB. Our algorithm, namely $\texttt{ALG}^{Rs}$, works as follows: at every round $t \in [T]$, it samples a number $x_t$ in $\left[-1 + \ln \frac{UCB_{\bar{h}_t, t}}{LCB_{\underline{h}_t, t}}, \ln \frac{UCB_{\bar{h}_t, t}}{LCB_{\underline{h}_t, t}}\right]$, uniformly at random. $\texttt{ALG}^{Rs}$ sends the packet $f_t$ s.t.

$$f_t \in \underset{\substack{f \in \mathcal{B}_t \text{ s.t.} \\ UCB_{f,t} \ge e^{x_t} LCB_{\underline{h}_t, t}}}{\arg\min} d_f, \quad (3)$$

*i.e.* the earliest-deadline packet s.t. its UCB is at least $e^{x_t}$ times the LCB of the heaviest packet. This procedure is summarized in Algorithm 5.

---

**Algorithm 5** $\text{ALG}^{Rs}$

---
1: **for** $t \in [T]$ **do**
2:     Compute UCBs and LCBs for every packet type, obtaining $\{UCB_{i,t}\}_{i \in [K]}$ and $\{LCB_{i,t}\}_{i \in [K]}$
3:     Receive new packets
4:     Identify $\underline{h}_t \in \arg\max_{h \in \mathcal{B}_t} LCB_{h,t}$ and $\bar{h}_t \in \arg\max_{h \in \mathcal{B}_t} UCB_{h,t}$.
5:     Sample a number $x_t$ from $\left[-1 + \ln \frac{UCB_{\bar{h}_t,t}}{LCB_{\underline{h}_t,t}}, \ln \frac{UCB_{\bar{h}_t,t}}{LCB_{\underline{h}_t,t}}\right]$
6:     Schedule $f_t$ according to Equation (3)
7: **end for**

---

**Theorem 5.2** (Upper Bound for the $\frac{e}{e-1}$-regret of $\text{ALG}^{Rs}$). *In $s$-bounded $K$-OPSD instances where randomization is allowed, $\text{ALG}^{Rs}$ satisfies*

$$\mathbb{E}[G_{OPT}] \le \frac{e}{e-1}\mathbb{E}[G_{\text{ALG}^{Rs}}] + \widetilde{\mathcal{O}}\left(\sqrt{KT}\right), \qquad (4)$$

*for every $s > 1$.*

The quantity $\frac{e}{e-1}$ is bigger than $\frac{5}{4}$, being the lower bound for the competitive ratio for randomized algorithms. However, it is an open question whether there exists an algorithm achieving a smaller-than-$\frac{e}{e-1}$ upper bound for any $s$-bounded instance, with arbitrary $s$. Our result matches the best existing upper bound while including learning, which poses an additional difficulty. Future advancements in the packet scheduling literature may lead to algorithms with sharper upper bounds on the competitive ratio, paving the way for their learning counterparts. Note that, while deterministic algorithms can achieve $\alpha$-no-regret using the optimism principle, randomized algorithms require a more complex estimation procedure. The involvement of randomization based on the ratio between unknown quantities leads to the usage of LCBs, together with UCBs.

## 6 Final Remarks and Future Directions

This is the first work exploring the interplay between bandit learning and online packet scheduling with deadlines. We introduce the $K$-OPSD problem, a theoretical framework that is motivated by applications such as QoS buffer management and digital advertising. We show that $K$-OPSD strictly generalizes the $K$-armed sleeping bandit problem, and develop a theory that naturally extends existing algorithms from sleeping bandits to $K$-OPSD.

There are several criticalities that emerge when trying to combine techniques from the toolbox of regret minimization in MABs and the analysis techniques for competitive ratio.

**Buffer Divergence between ALG and OPT** In MABs, an algorithm can always choose the same action as OPT. This is not the case in $K$-OPSD. A single different decision leads to different buffers in the subsequent rounds. Thus, it is not possible, in general, to bound the instantaneous regret of

an algorithm in this setting. Bounding the $\alpha$-regret in $K$-OPSD requires a charging scheme as an intermediate step that maps a set of subsequent actions from the algorithm to a set of actions in OPT schedule. This issue is particularly critical in randomized algorithms, where the stochastic structure exhibits an interplay between the randomness of the environment and the one of the algorithm.

**Breaking the $\Phi$ barrier for the competitive ratio** In the 2-bounded OPSD setting, when the number of distinct weights is finite and equal to $K$, we provide an algorithm that achieves a competitive ratio of $\theta_K < \Phi$. The values of $\theta_K$ span from $\sqrt{2}$ to $\Phi$ as $K$ grows from 2 to $\infty$. The algorithm builds over the well-known family of $\beta$-Earliest Deadline First ($\beta$-EDF) algorithms. $\beta$-EDF uses a static threshold to decide how close to the expiration the next packet to schedule must be. Our algorithm, $\text{ALG}^\theta$, uses a dynamic threshold and a restart mechanism. The sequence of thresholds is the solution to a system of $K + 1$ equations that converge to the lower bound instance from (Hajek, 2001). Using a dynamic threshold requires a carefully crafted charging mechanism, where the schedule of $\text{ALG}^\theta$ is divided in epochs. We write the competitive ratio as the solution to a linear optimization problem, and search for the worst possible vertex of the solution space.

**Providing a $\theta_{K-1}$-regret lower bound** We provide a lower bound for the $\theta_{K-1}$-regret in 2-bounded $K$-OPSD instances of $\Omega\left(\sqrt{T}\right)$. Proving a lower bound on the regret in bandits usually requires two instances that are hard to distinguish (from a statistical perspective) but with different enough optimal strategies. On the other hand, proving a lower bound on the competitive ratio for OPSD requires a carefully crafted instance where every decision always leads to the same competitive ratio. The two objectives can easily collide: two instances where every decision always leads to the same outcome cannot be different enough to generate a meaningful regret lower bound. We overcome this problem by proposing a *three-instance* construction where the arrival sequence is partitioned into gadgets that repeats.

**UCBs are not enough for randomized algorithms** Our algorithms highlight a crucial difference in the construction of deterministic learning algorithms from randomized ones. Deterministic learning algorithms only use the notion of *upper confidence bound* (UCB) to estimate the expected value of a packet type. Being optimistic is enough to upper bound the $\alpha$-regret in this case. Instead, our randomized algorithms make use of both UCBs and lower confidence bounds (LCBs). Indeed, LCBs allow randomized algorithms to deal with ratios of estimates in a more natural way. This emerges from the proofs of Theorems 5.1 and 5.2.

## 6.1 Future Directions

Many interesting future directions can be explored. First, for deterministic algorithms, providing tight $\Phi$-regret for general instances is a challenging problem. Also, combining both randomized algorithms and the $K$-finite type assumptions can be explored to have competitive ratios even closer to 1. We conjecture that the competitive ratio can be lowered below $\frac{5}{4}$, up to $\frac{7}{6} = 1.1\bar{6}$ when $K = 2$, employing a similar lower-bound matching algorithm over the construction Theorem 4 of (Chin & Fung, 2003). Finally, additional inter-packet structure can be further explored, developing models resembling linear bandits and/or contextual bandits. All these steps towards generalization are interesting and technically challenging, but can be well-motivated by real-world applications.

## Acknowledgements

VP's research was supported in part by the French National Research Agency (ANR) in the framework of the PEPR IA FOUNDRY project (ANR-23-PEIA-0003) and through the grant DOOM ANR-23-CE23-0002. It was also funded by the European Union (ERC, Ocean, 101071601).

## Impact Statement

This paper presents work whose goal is to advance the field of Machine Learning. There are many potential societal consequences of our work, none which we feel must be specifically highlighted here.

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

# A    Technical Lemmas

**Lemma A.1** ((Levy et al., 2024)). *Let $\delta \in (0,1)$. Let $t_{i,n} \in [T]$ be the time at which packet type $i \in [K]$ is selected for the $n$-th time. Let $\mathcal{E}(\delta)$ be the good event under which:*

- $0 < LCB_{i,t_{i,1}} < \ldots < LCB_{i,t_{i,n}} \leq \mu_i$ *for every $i \in [K]$ and $n \in [T]$.*

- $\mu_i \leq UCB_{i,t_{i,n}} < \ldots < UCB_{i,t_{i,1}} = 1$ *for every $i \in [K]$ and $n \in [T]$.*

- $UCB_{i,t_{i,n}} - LCB_{i,t_{i,n}} \leq 2\beta_{i,t_{i,n}}$ *for every $i \in [K]$ and $n \in [T]$.*

*Then, we have $\mathbb{P}(\mathcal{E}(\delta)) \geq 1 - \delta$.*

**Lemma A.2.** *Let $i, j \in [K]$ be packet types with means $\mu_i > \mu_j$ and gap $\Delta_{i,j} := \mu_i - \mu_j > 0$. Then, on the event $\mathcal{E}\left(\frac{1}{T}\right)$, the event*

$$\left\{ \mathrm{UCB}_{j,t} > \mathrm{UCB}_{i,t} \right\} \cap \left\{ i, j \in \mathcal{B}_t \right\} \cap \left\{ j \text{ is scheduled at time } t \right\}$$

*can occur at most*

$$Q_{i,j} = \frac{8 \ln T}{\Delta_{i,j}^2}$$

*times.*

*Proof.* We work under the event $\mathcal{E}\left(\frac{1}{T}\right)$. Fix any round $t$ for which $i, j \in \mathcal{B}_t$ and arm $j$ is scheduled. Consider $\mathrm{UCB}_{j,t} > \mathrm{UCB}_{i,t}$. Unfolding the indices and adding and subtracting the true means,

$$\widehat{\mu}_{j,t} + \sqrt{\frac{2 \ln T}{N_{j,t}}} > \widehat{\mu}_{i,t} + \sqrt{\frac{2 \ln T}{N_{i,t}}} \implies \left(\mu_j - (\mu_j - \widehat{\mu}_{j,t})\right) + \sqrt{\frac{2 \ln T}{N_{j,t}}}$$
$$> \left(\mu_i - (\mu_i - \widehat{\mu}_{i,t})\right) + \sqrt{\frac{2 \ln T}{N_{i,t}}}.$$

Hence $(\mu_j - \widehat{\mu}_{j,t}) \leq \sqrt{2 \ln T / N_{j,t}}$ and $(\mu_i - \widehat{\mu}_{i,t}) \geq -\sqrt{2 \ln T / N_{i,t}}$, so the previous display yields

$$\mu_j - \sqrt{\frac{2 \ln T}{N_{j,t}}} + \sqrt{\frac{2 \ln T}{N_{j,t}}} > \mu_i - \sqrt{\frac{2 \ln T}{N_{i,t}}} + \sqrt{\frac{2 \ln T}{N_{i,t}}},$$

that is,

$$\mu_i - \mu_j \leq 2\sqrt{\frac{2 \ln T}{N_{j,t}}}.$$

Equivalently,

$$N_{j,t} \leq \frac{8 \ln T}{\Delta_{i,j}^2}. \tag{5}$$

Now observe that each time the event

$$\left\{ \mathrm{UCB}_{j,t} > \mathrm{UCB}_{i,t} \right\} \cap \left\{ i, j \in \mathcal{B}_t \right\} \cap \left\{ j \text{ is scheduled} \right\}$$

occurs, the counter $N_{j,t}$ increases by one. Thus, if this event were to occur more than $m$ times, then there would exist a round $t$ with $N_{j,t} > m$ at which it still occurs. Choosing any $m \geq \dfrac{8 \ln T}{\Delta_{i,j}^2}$ contradicts (5). In particular, taking

$$m = \frac{8 \ln T}{\Delta_{i,j}^2}$$

is valid (it is a looser threshold), and therefore the event in the statement can occur at most $Q_{i,j} = 8 \ln T / \Delta_{i,j}^2$ times. $\qquad\square$

The following lemma includes some properties of $\widehat{p}_{ab,t}$, which are crucial for the analysis of $\mathtt{ALG}^{R2}$. Let $\mathcal{E}(\delta)$ be the *good event* under which all confidence intervals hold simultaneously for every type $i \in [K]$ and round $t \in [T]$.

**Lemma A.3** (Properties of $\widehat{p}_{a_t b_t, t}$). *Let $a_t^* \in \arg\max_{a \in V_t^*} w_a$ and $b_t^* \in \arg\max_{b \in B_t^*} w_b$, where $V_t^*$ is the set of pending packets with deadline $t$ for $\mathtt{OPT}$ at time $t$, and $B_t^*$ is the set of the ones having deadline greater than $t$. Then, under the good event $\mathcal{E}_T(\delta)$, the followings hold for every $t \in [T]$:*

$$5\widehat{p}_{a_t b_t, t} a_t \geq 4a_t^* - b_t^* - 10\beta_{a_t, t} \tag{6a}$$

$$5(\widehat{p}_{a_t b_t, t} a_t + \widehat{q}_{a_t b_t, t} b_t) \geq 4b_t^* - 10\beta_{a_t, t} - 8\beta_{b_t, t} \tag{6b}$$

$$5\widehat{p}_{a_t b_t, t} a_t + 2\widehat{q}_{a_t b_t, t} b_t \geq 4a_t^* - 10\beta_{a_t, t} \tag{6c}$$

$$5\widehat{p}_{a_t b_t, t} a_t + 2\widehat{q}_{a_t b_t, t} b_t \geq b_t^* - 10\beta_{a_t, t} - 2\beta_{b_t, t} \tag{6d}$$

*Proof.* Under the *good event*, all confidence intervals contain the true mean. Thus $LCB_{i,t} \leq \mu_i \leq UCB_{i,t}$ and $LCB_{i,t} + 2\beta_{i,t} \geq \mu_i \geq UCB_{i,t} - 2\beta_{i,t}$ for every $i \in [K]$ and $t \in [T]$.

Proof of Equation (6a)

$$5\widehat{p}_{a_t b_t, t} a_t \geq 5\widehat{p}_{a_t b_t, t} UCB_{a_t, t} - 10\widehat{p}_{a_t b_t, t} \beta_{a_t, t}$$
$$\geq 4UCB_{a_t, t} - LCB_{b_t, t} - 10\beta_{a_t, t}$$
$$\geq 4UCB_{a_t^*, t} - LCB_{b_t^*, t} - 10\beta_{a_t, t}$$
$$\geq 4a_t^* - b_t^* - 10\beta_{a_t, t}.$$

Proof of Equation (6b)

$$5(\widehat{p}_{a_t b_t, t} a_t + \widehat{q}_{a_t b_t, t} b_t) \geq 5(\widehat{p}_{a_t b_t, t} UCB_{a_t, t} + \widehat{q}_{a_t b_t, t} LCB_{b_t, t}) - 10\widehat{p}_{a_t b_t, t} \beta_{a_t, t}$$
$$\geq 5(\widehat{p}_{a_t b_t, t} UCB_{a_t, t} + \widehat{q}_{a_t b_t, t} LCB_{b_t, t}) - 10\widehat{p}_{a_t b_t, t} \beta_{a_t, t}$$
$$\geq 4LCB_{b_t, t} - 10\beta_{a_t, t}$$
$$\geq 4UCB_{b_t, t} - 10\beta_{a_t, t} - 8\beta_{b_t, t}$$
$$\geq 4b_t^* - 10\beta_{a_t, t} - 8\beta_{b_t, t}.$$

Proof of Equation (6c)

$$5\widehat{p}_{a_t b_t, t} a_t + 2\widehat{q}_{a_t b_t, t} b_t \geq 5\widehat{p}_{a_t b_t, t} UCB_{a_t, t} + 2\widehat{q}_{a_t b_t, t} LCB_{b_t, t} - 10\widehat{p}_{a_t b_t, t} \beta_{a_t, t}$$
$$\geq 4UCB_{a_t, t} - 10\widehat{p}_{a_t b_t, t} \beta_{a_t, t}$$
$$\geq 4a_t^* - 10\beta_{a_t, t}.$$

Proof of Equation (6d)

$$5\widehat{p}_{a_t b_t, t} a_t + 2\widehat{q}_{a_t b_t, t} b_t \geq 5\widehat{p}_{a_t b_t, t} UCB_{a_t, t} + 2\widehat{q}_{a_t b_t, t} LCB_{b_t, t} - 10\widehat{p}_{a_t b_t, t} \beta_{a_t, t}$$
$$\geq LCB_{b_t, t} - 10\widehat{p}_{a_t b_t, t} \beta_{a_t, t}$$
$$\geq UCB_{b_t, t} - 10\beta_{a_t, t} - 2\beta_{b_t, t}$$
$$\geq b_t^* - 10\beta_{a_t, t} - 2\beta_{b_t, t}.$$

$\square$

# B    Omitted Proofs

**Lemma 2.1** (Sleeping Bandits are a special case of $K$-OPSD). *Every 1-bounded instance of the $K$-OPSD problem can be reduced to a $K$-armed Sleeping Bandit problem, and vice versa.*

*Proof.* By the definition of $K$-armed sleeping bandits of (Kleinberg et al., 2010), there exist $K$ types of actions and at every round $t \in [T]$ an adversary selects a subset $\mathcal{A}_t \subseteq [K]$ to propose to the learner. Actions selected in time $t$ are only available in that round, and unless the adversary selects them again, are not available anymore after $t$, since at most one action can be played per round. In 1-bounded $K$-OPSD, we can just ignore packets with the same type except for one: the packets with the same type are indistinguishable and only one of them can eventually be played. For every action $a \in \mathcal{A}_t$, we create a buffer $\mathcal{B}_t$ s.t. there exists at least one packet $p_a$ of type $a \in \mathcal{B}_t$. Then, no matter how many copies of $p_a$ there are in the buffer, the decision space is always the same and is equivalent to the size of the minimal buffer, which is actually $\mathcal{A}_t$. $\square$

## B.1 Deterministic Algorithms

**Lemma 3.1** (Sleeping Bandits are a special case of $K$-OPSD)**.** *Every* 1*-bounded instance of the $K$-OPSD problem can be reduced to a $K$-armed Sleeping Bandit problem, and viceversa.*

*Proof.* By the definition of $K$-armed sleeping bandits of (Kleinberg et al., 2010), there exist $K$ types of actions and at every round $t \in [T]$ an adversary selects a subset $\mathcal{A}_t \subseteq [K]$ to propose to the learner. Actions selected in time $t$ are only available in that round, and unless the adversary selects them again, are not available anymore after $t$, since at most one action can be played per round. In 1-bounded $K$-OPSD, we can just ignore packets with the same type except for one: the packets with the same type are indistinguishable and only one of them can eventually be played. For every action $a \in \mathcal{A}_t$, we create a buffer $\mathcal{B}_t$ s.t. there exists at least one packet $p_a$ of type $a \in \mathcal{B}_t$. Then, no matter how many copies of $p_a$ there are in the buffer, the decision space is always the same and is equivalent to the size of the minimal buffer, which is actually $\mathcal{A}_t$. $\square$

**Proposition 3.1** ($\Phi$-regret of $\text{EDF}_\Phi^L$)**.** *In* 2*-bounded and* 3*-bounded $K$-OPSD instances, $\text{EDF}_\Phi^L$ suffers an expected cumulative $\Phi$-regret bounded as*

$$R_{\Phi,T}^{\text{EDF}_\Phi^L} \le \mathcal{O}\left( K \sum_{i=1}^{K-1} \frac{\ln T}{\Delta_{i,i+1}} + \sum_{\substack{i,j \in [K] \\ \Delta_{i,j}^\Phi > 0}} \frac{\ln T}{\Delta_{i,j}^\Phi} \right),$$

*where $\Delta_{i,j}^\Phi \coloneqq |\mu_i - \Phi\mu_j|$. Alternatively, the $\Phi$-regret can be bounded as*

$$R_{\Phi,T}^{\text{EDF}_\Phi^L} \le \widetilde{\mathcal{O}}\left( \sqrt{KT} \right).$$

*Proof.* We call $f_t$ the packet sent by $\text{ALG}$ in $t$, and $j_t$ the packet sent by $\text{OPT}$ in $t$. We also call $h_t$ the packet in the buffer of $\text{ALG}$ at time $t$ such that $h_t \in \arg\max_{h \in \mathcal{B}_t} UCB_{h,t}$. To lighten the notation, we let the name of the packet coincide with its type.

We consider the *charging scheme* in which $j_t$ is charged to its copy in the $\text{ALG}$ schedule up to $t$ if it exists, otherwise it is charged to $f_t$.

For $\delta \in (0, 1)$, define the uniform "good" event

$$\mathcal{E}(\delta) \coloneqq \bigcap_{k=1}^K \bigcap_{n=1}^T \left\{ |\widehat{\mu}_{k,n} - \mu_k| \le \sqrt{\frac{2 \ln \delta^{-1}}{n}} \right\},$$

where $\widehat{\mu}_{k,n}$ denotes the empirical mean of arm $k$ after $n$ samples. Lemma A.1 states that the good event holds with probability at least $1 - \delta$. Let $\delta = \frac{1}{T}$. Since

$$\mathbb{E}[R_{\Phi,T}^{\text{EDF}_\Phi^L}] = \mathbb{E}\left[ R_{\Phi,T}^{\text{EDF}_\Phi^L} | \mathcal{E}(\delta) \right] \mathbb{P}\left( \mathcal{E}(\delta) \right) + \mathbb{E}\left[ R_{\Phi,T}^{\text{EDF}_\Phi^L} | \mathcal{E}(\delta)^C \right] \mathbb{P}\left( \mathcal{E}(\delta)^C \right)$$

$$\le \mathbb{E}\left[ R_{\Phi,T}^{\text{EDF}_\Phi^L} | \mathcal{E}(\delta) \right] \mathbb{P}\left( \mathcal{E}(\delta) \right) + \mathcal{O}(1),$$

we focus on bounding the first addendum under the good event.

We proceed by cases.

**Case 1: $f_t$ only receives one charge**  Ideally, we want to show that $f_t$ receives a charge that is at most $\Phi f_t$.

**Case 1a: $f_t$ receives a charge from $f_t$**  Then the total charge is obviously $f_t$. No $\Phi$-regret is accrued in this case.

**Case 1b: $f_t$ receives a charge from $j_t$**  Then, $\text{ALG}$ hasn't send $j_t$ before $f_t$. This implies that $j_t \in \mathcal{B}_t$ and that $\Phi UCB_{j_t,t} < UCB_{h_t,t} \le \Phi UCB_{f_t,t}$, and $UCB_{j_t,t} < UCB_{f_t,t}$. If $j_t \le f_t$, then the total charge is trivially bounded by $f_t$. If $j_t > f_t$, the instantaneous $\Phi$-regret can be bounded as

$$j_t - \Phi f_t < UCB_{j_t,t} - \Phi f_t < \Phi UCB_{f_t,t} - \Phi f_t \le \Phi \beta_{f_t,t}.$$

Since $UCB_{j_t,t} < UCB_{f_t,t}$ can occur at most $Q_{j_t,f_t} = \frac{8\ln T}{\Delta_{j_t,f_t}^2}$ times (Lemma A.2), the total $\Phi$-regret accrued in this case is bounded as

$$\Phi \sum_{t=1}^{Q_{j_t,f_t}} \beta_{f_t,t} \leq \frac{8\Phi \ln T}{\Delta_{j_t,f_t}}.$$

**Case 2:** $f_t$ **receives a charge from** $j_t$ **and a charge from** $f_t$    In this case, $j_t \in \mathcal{B}_t$ and OPT has scheduled $f_t$ after time $t$. Also, $\Phi UCB_{j_t,t} < UCB_{h_t,t} \leq \Phi UCB_{f_t,t}$. We have to distinguish two cases.

**Case 2a:** $d_{f_t} = t + 2$    In this case, we have $UCB_{f_t,t} = UCB_{h_t,t}$. If $\Phi j_t \leq f_t$, we have $j_t + f_t \leq (1 + \Phi^{-1}) f_t = \Phi f_t$, and no $\Phi$-regret is accrued. If $\Phi j_t > f_t$, we have that $\Phi UCB_{j_t,t} \leq UCB_{f_t,t}$ can occur at most $Q_{j_t,t}^{\Phi} = \frac{8\ln T}{(\Delta_{f_t,j_t}^{\Phi})^2}$ times (Lemma A.2), where $\Delta_{f_t,j_t}^{\Phi} > 0$. The instantaneous $\Phi$-regret can be bounded as

$$(j_t + f_t) - \Phi f_t < UCB_{j_t,t} - \Phi^{-1} f_t < \Phi^{-1} UCB_{f_t,t} - \Phi^{-1} f_t \leq \Phi^{-1} \beta_{f_t,t}.$$

The total $\Phi$-regret accrued in this case is bounded as

$$\Phi^{-1} \sum_{t=1}^{Q_{j_t,f_t}^{\Phi}} \beta_{f_t,t} \leq \frac{8\ln T}{\Delta_{f_t,j_t}^{\Phi}},$$

with $\Delta_{f_t,j_t}^{\Phi} > 0$.

**Case 2b:** $d_{f_t} < t + 2$    In this case, we have $d_{h_t} = t + 2$ and $d_{j_t} \leq d_{f_t}$, since the schedule of OPT is ordered with ascending deadlines. Since $f_t$ is played by OPT after $t$, and of course before $t + 2$, it must be that $d_{f_t} = t + 1$. So we consider the couple of packets scheduled by ALG in $t$ and $t + 1$, a bound the $\Phi$-regret accrued in these two rounds jointly. Note that $f_{t+1}$ can receive a charge only from its copy, since OPT plays $f_t$ in $t + 1$ and is charged to $f_t$ itself. Moreover, $\Phi UCB_{f_t,t} \geq UCB_{h_t,t}$. If $\Phi j_t < f_t$, we have $(j_t + f_t + f_{t+1}) - \Phi(f_t + f_{t+1}) < (\Phi^{-1} f_t + f_t + f_{t+1}) - \Phi(f_t + t_{t+1}) < 0$, and there is no $\Phi$-regret. If $\Phi j_t \geq f_t$, the instantaneous $\Phi$-regret can be bounded as

$$(j_t + f_t + f_{t+1}) - \Phi(f_t + f_{t+1}) < UCB_{j_t,t} - \Phi^{-1}(f_t + f_{t+1})$$
$$\leq \Phi^{-1} \frac{1}{2} 2 UCB_{h_t,t} - \Phi^{-1}(f_t + f_{t+1})$$
$$\leq \frac{1}{2\Phi} \Phi(UCB_{f_t,t} + UCB_{f_{t+1},t}) - \Phi^{-1}(f_t + f_{t+1})$$
$$\leq \frac{3}{2}(\beta_{f_t,t} + \beta_{f_{t+1},t}).$$

Since $j_t \in \mathcal{B}$, this situation can occur until $UCB_{j_t,t} < UCB_{f_t,t}$, thus at most $Q_{j_t,f_t}$ times (Lemma A.2). The total $\Phi$-regret accrued in this case is bounded as

$$\frac{3}{2} \sum_{t=1}^{Q_{j_t,f_t}} \beta_{f_t,t} \leq \frac{8\Phi \ln T}{\Delta_{j_t,f_t}}.$$

Summing up all the possible regrets sources for every round, and recombining by packet type, yields the first result:

$$R_{\Phi,T}^{\text{EDF}_{\Phi}^L} \leq \mathcal{O}\left(\sum_{t=1}^{T}\left(\frac{1}{\Delta_{j_t,f_t}} + \frac{1}{\Delta_{f_t,j_t}^{\Phi}}\right)\ln T\right)$$
$$\leq \mathcal{O}\left(K \sum_{i=1}^{K-1} \frac{\ln T}{\Delta_{i,i+1}} + \sum_{\substack{i,j\in[K] \\ \Delta_{i,j}^{\Phi}>0}} \frac{\ln T}{\Delta_{i,j}^{\Phi}}\right).$$

The second bound can be obtained by noting that, every time a packet $f_t$ is played, the instantaneous $\Phi$-regret at round $t$ can be bounded by $\beta_{f_t,t}$. Summing up for every packet type $i \in [K]$, and for the number of times a packet of type $i$ has been scheduled, we get

$$\sum_{i\in[K]} \sum_{n=1}^{N_{i,T}} \beta_{i,t_{i,n}} \leq \widetilde{\mathcal{O}}\left(\sum_{i\in[K]} \sqrt{N_{i,T}}\right) \leq \widetilde{\mathcal{O}}\left(\sqrt{KT}\right),$$

where $t_{i,n}$ is the round in which a packet of type $i$ has been scheduled for the $n$-th time.                                                   □

**Proposition 4.1** ($\theta_K$- Competitive Ratio for $\text{ALG}^\theta$)**.** *In 2-bounded $K$-OPSD instances, $\text{ALG}^\theta$ satisfies*

$$G_{OPT} \le \theta_K G_{\text{ALG}^\theta}.$$

*Proof.* We analyze separately the two types of possible epochs, *i.e.* the ones ending with $b_{t+\ell}$ (*first type*) and the ones ending with $v_{t+\ell}$ (*second type*), where $\ell$ will be used to indicate the length of an epoch. Note that, by construction, any epoch can be at most $K$ rounds long, *i.e.* $\ell \le K$. We have to show two things: (1) every epoch starts with the buffer of $\text{OPT}$ being entirely contained in the buffer of $\text{ALG}^\theta$, thus $\text{OPT}$ has no advantage gained from previous epochs; (2) $G_{\text{OPT}}^E \le \theta_K G_{\text{ALG}^\theta}^E$ for every epoch $E$, where $G_{\text{OPT}}^E$ indicates the gain of $\text{OPT}$ inside epoch $E$, and $G_{\text{ALG}^\theta}^E$ indicates the gain of $\text{ALG}^\theta$ inside epoch $E$. These two facts combined yield a global competitive ratio of $\theta_K$.

**(1) $\text{ALG}^\theta$ starts every epoch with a larger or equal buffer to OPT.** We prove this fact by allowing for a *stronger* $\text{OPT}$. In particular, at the end of every epoch, we allow $\text{OPT}$ to send additional packets until its buffer is entirely contained in the buffer of $\text{ALG}^\theta$.

**(1a) $\{v_t, \ldots, b_{t+\ell}\}$-type epochs.** Consider an epoch $E$ starting in $t$ and ending in $t + \ell$ of the first type, then we have:

$$G_{\text{ALG}^\theta}^E = \sum_{j=0}^{\ell-1} v_{t+j} + b_{t+\ell}.$$

Assuming that $\text{OPT}$ starts with a buffer that is smaller of equal than the one of $\text{ALG}^\theta$, we upper bound its gain inside epoch $E$ as

$$G_{\text{OPT}}^E \le \sum_{j=0}^{\ell} b_{t+j} + v_{t+\ell}, \tag{7}$$

which is the best possible gain inside epoch $E$ since $b_{t+j} > v_{t+j}$ for every $j < \ell$ by construction, $v_{t+\ell} + b_{t+\ell} \ge b_{t+\ell}$. We allowed $\text{OPT}$ to schedule an extra packet w.r.t. to $\text{ALG}^\theta$, which is $v_{t+\ell}$. This way, when epoch $E$ terminates, the following epoch starts with the buffer of $\text{OPT}$ contained in the one of $\text{ALG}^\theta$.

**(2a) $\{v_t, \ldots, v_{t+\ell}\}$-type epochs** Consider an epoch $E$ starting in $t$ and ending in $t + \ell$ of the second type, then we have:

$$G_{\text{ALG}^\theta}^E = \sum_{j=0}^{\ell} v_{t+j}.$$

Assuming that $\text{OPT}$ starts with a buffer that is smaller of equal than the one of $\text{ALG}^\theta$, we upper bound its gain inside epoch $E$ as

$$G_{\text{OPT}}^E \le \sum_{j=0}^{\ell-1} b_{t+j} + v_{t+\ell}, \tag{8}$$

which is the best possible gain inside epoch $E$ since $b_{t+j} > v_{t+j}$ for every $j < \ell$ by construction, and $v_{t+\ell} \ge b_{t+\ell}$. When epoch $E$ terminates, the following epoch starts with the buffer of $\text{OPT}$ contained in the one of $\text{ALG}^\theta$.

**(2) Bounding the competitive ratio of $\text{ALG}^\theta$ in every epoch.** We now prove that, given an epoch $E$ of any type, $G_{\text{OPT}} \le \theta_K G_{\text{ALG}^\theta}^E$.

**(2a) $\{v_t, \ldots, b_{t+\ell}\}$-type epochs.** Consider an epoch $E$ starting in $t$ and ending in $t + \ell$ of the first type, then we have:

$$\theta_K G_{\text{ALG}^\theta}^E - G_{\text{OPT}}^E \ge \theta_K \sum_{j=0}^{\ell-1} v_{t+j} + \theta_K b_{t+\ell} - \sum_{j=0}^{\ell} b_{t+j} - v_{t+\ell}, \tag{9}$$

which follows by Equation (7). Our goal is to prove that the RHS of Equation (9) is nonnegative for any choice of the sequences $\{v_{t+j}\}_{j=0}^\ell$ and $\{b_{t+j}\}_{j=0}^\ell$. The epoch termination conditions of $\text{ALG}^\theta$, together with the definitions of $v_{t+j}$ and

$b_{t+j}$, impose a set of constraints on these sequences:

$$v_{t+j} \geq b_{t+j-1}, \quad 1 \leq j \leq K-1, \tag{10}$$

$$b_{t+j} \geq v_{t+j}, \quad j \geq 0, \tag{11}$$

$$v_{t+j} \geq \frac{x_j}{x_{j+1}} b_{t+j}, \quad 0 \leq j \leq \ell-1, \tag{12}$$

$$v_{t+\ell} \leq \frac{x_\ell}{x_{\ell+1}} b_{t+\ell}, \tag{13}$$

where (10) follows from the definition of $v_{t+j}$, (11) follows from the epoch termination conditions of $\mathtt{ALG}^\theta$, and (12) and (13) follow from the decision rule of $\mathtt{ALG}^\theta$ (and with the notational convention that $x_0 = 1$ and $x_K = x_{K-1}$). An important consequence of these conditions is that both $v_{t+j}$ and $b_{t+j}$ are strictly increasing sequences.

Intuitively, to prove that under these constraints the RHS of Equation (9) is nonnegative, we look at it from the perspective of a constrained minimization problem, and show that the feasible minimum set the RHS to 0. In particular, note that both the objective function (9) and the constraints (10)-(13) are linear in the parameters $v_{t+j}$ and $b_{t+j}$. Thus, the minimum must be in one of the edges of the decision space defined by the constraints. We now define an edge sequence $\widetilde{v}_{t+j}$ and $\widetilde{b}_{t+j}$ where inequalities are tight and the RHS of (9) is 0. Let $s > 0$ be any scale parameter (the problem is homogeneous and the sign does not change with rescaling). Then, set

$$\widetilde{b}_t = s, \tag{14}$$

$$\widetilde{v}_t = s(\theta_K - 1), \tag{15}$$

$$\widetilde{v}_{t+j} = \widetilde{b}_{t+j-1}, \quad j \geq 1, \tag{16}$$

$$\widetilde{v}_{t+j} = \frac{x_j}{x_{j+1}} \widetilde{b}_{t+j}, \quad 0 \leq j \leq \ell. \tag{17}$$

A direct implication of (14)-(17) is that

$$\widetilde{v}_{t+j} = \widetilde{b}_{t+j-1} = \frac{x_j}{x_{j-1}} \widetilde{v}_{t+j-1} = \frac{x_j}{x_0} \widetilde{v}_t = s(\theta_K - 1)x_j,$$

$$\widetilde{b}_{t+j} = \frac{x_{j+1}}{x_j} \widetilde{v}_{t+j} = \frac{x_{j+1}}{x_0} \widetilde{v}_t = s(\theta_K - 1)x_{j+1}.$$

We rearrange (9), plug the edge sequence and write

$$\theta_K \sum_{j=0}^{\ell-1} \widetilde{v}_{t+j} + \theta_K \widetilde{b}_{t+\ell} - \sum_{j=0}^{\ell} \widetilde{b}_{t+j} - \widetilde{v}_{t+\ell} = \sum_{j=0}^{\ell-1} (\theta_K \widetilde{v}_{t+j} - \widetilde{b}_{t+j}) + (\theta_K - 1)\widetilde{b}_{t+\ell} - \widetilde{v}_{t+\ell}$$

$$= s(\theta_K - 1)\left[\theta_K(1 + \ldots + x_{\ell-1} + x_{\ell+1}) - (x_1 + \ldots + 2x_\ell + x_{\ell+1})\right] = 0,$$

which is a direct consequence of the definition of $\{x_j\}_{j=0}^{K-1}$ as a solution to (1). We proved that the sequences $\{\widetilde{v}_{t+j}\}_{j=0}^{\ell}$ and $\{\widetilde{b}_{t+j}\}_{j=0}^{\ell}$ are feasible and make all the constraints tight. This step of the proof can be concluded by noting that these edge sequences are actually the minimizer of the RHS of (9): the objective is linear with a positive coefficient in $\{v_{t+j}\}_{j=0}^{\ell-1}$ and $b_{t+\ell}$, and we set the minimum possible values for those (all lower bounds became equalities); the objective is linear with a negative coefficient in $\{b_{t+j}\}_{j=0}^{\ell-1}$ and $v_{t+\ell}$, and we set the maximum possible values for those (all upper bounds became equalities). Thus, the term cannot be reduced anymore and it is nonnegative for any feasible sequence of values, and $G_{\mathtt{OPT}}^E \leq \theta_K G_{\mathtt{ALG}^\theta}^E$.

**(2b)** $\{v_t, \ldots, v_{t+\ell}\}$**-type epochs.** We closely follow the argument developed in the previous step. Consider an epoch $E$ starting in $t$ and ending in $t + \ell$ of the first type, then we have:

$$\theta_K G_{\mathtt{ALG}^\theta}^E - G_{\mathtt{OPT}}^E \geq \theta_K \sum_{j=0}^{\ell} v_{t+j} - \sum_{j=0}^{\ell-1} b_{t+j} - v_{t+\ell}, \tag{18}$$

which follows by Equation (8). Our goal is to prove that the RHS of Equation (18) is nonnegative for any choice of the sequences $\{v_{t+j}\}_{j=0}^{\ell}$ and $\{b_{t+j}\}_{j=0}^{\ell}$. Constraints (10)-(12) still hold, in addition we have

$$v_{t+\ell} \geq b_{t+\ell,}, \tag{19}$$

which follows from the epoch termination condition of $\text{ALG}^{\theta}$. As before, we need to prove that the RHS of (18) is nonnegative under any sequence satisfying the constraints.

We use the same constrained optimization argument as before, and define and edge sequence which makes the constraints tight:

$$\widetilde{b}_t = s, \tag{20}$$

$$\widetilde{v}_t = s(\theta_K - 1), \tag{21}$$

$$\widetilde{v}_{t+j} = \widetilde{b}_{t+j-1}, \quad j \geq 1, \tag{22}$$

$$\widetilde{v}_{t+j} = \frac{x_j}{x_{j+1}} \widetilde{b}_{t+j}, \quad 0 \leq j \leq \ell - 1, \tag{23}$$

$$\widetilde{v}_{t+\ell} = \widetilde{b}_{t+\ell,}. \tag{24}$$

We now rewrite the RHS of (18) and plug the edge sequences to obtain:

$$\theta_K \sum_{j=0}^{\ell} \widetilde{v}_{t+j} - \sum_{j=0}^{\ell-1} \widetilde{b}_{t+j} - \widetilde{v}_{t+\ell} = \sum_{j=0}^{\ell-1} (\theta_K \widetilde{v}_{t+j} - \widetilde{b}_{t+j}) + (\theta_K - 1)\widetilde{v}_{t+\ell}$$

$$= s(\theta_K - 1)\left[\theta_K(1 + \ldots + x_\ell) - (x_1 + \ldots + 2x_\ell)\right] \geq 0,$$

which is a direct consequence of the definition of $\{x_j\}_{j=0}^{K-1}$ as a solution to (1). We proved that the sequences $\{\widetilde{v}_{t+j}\}_{j=0}^{\ell}$ and $\{\widetilde{b}_{t+j}\}_{j=0}^{\ell}$ are feasible and make all the constraints tight. This step of the proof can be concluded by noting that these edge sequences are actually the minimizer of the RHS of (18): the objective is linear with a positive coefficient in $\{v_{t+j}\}_{j=0}^{\ell}$, and we set the minimum possible values for those (all lower bounds became equalities); the objective is linear with a negative coefficient in $\{b_{t+j}\}_{j=0}^{\ell-1}$, and we set the maximum possible values for those (all upper bounds became equalities). Thus, the term cannot be reduced anymore, and it is nonnegative for any feasible sequence of values, and $G_{\text{OPT}}^E \leq \theta_K G_{\text{ALG}^{\theta}}^E$.

Putting all together completes the proof. □

**Theorem 4.2** (Upper Bound on the $\theta_K$-regret of $\text{ALG}^{\theta,U}$)**.** *For every instance of the stochastic $K$-types packet scheduling problem, we have*

$$\mathbb{E}[G_{\text{OPT}}] \leq \theta_K \mathbb{E}[G_{\text{ALG}^{\theta,U}}] + \widetilde{\mathcal{O}}\left(\sqrt{KT}\right).$$

*Proof.* The proof follows the argument of the previous one with the addition of optimism. We work under the good event $\mathcal{E}(\delta)$. Due to $\text{ALG}^{\theta,U}$ epoch termination conditions, all the packets in OPT buffer at the start of an epoch must also be contained in $\text{ALG}^{\theta,U}$ buffer. We already proved this in the proof of Proposition 4, and the same argument still holds. What we prove now is that the $\theta_K$-regret accrued inside an epoch is bounded by the sum of the widths of the confidence intervals of the packets selected by $\text{ALG}^{\theta,U}$.

**(a) $\{v_t, \ldots, b_{t+\ell}\}$-type epochs.** Consider an epoch $E$ starting in $t$ and ending in $t + \ell$ of the first type. Then

$$\mathbb{E}[G_{\text{OPT}}^E] \leq \sum_{j=0}^{\ell} UCB_{b_{t+j},t} + UCB_{v_{t+\ell},t},$$

which is an overestimation of the total reward that any algorithm can achieve within the epoch, due to epoch termination conditions of $\text{ALG}^{\theta,U}$. Note that under the good event, all confidence intervals contain the true mean, for every packet type

and time $t \in [T]$. We want to lower bound the following:

$$
\begin{aligned}
\theta_K \mathbb{E}[G^E_{\text{ALG}^{\theta,U}}] - \mathbb{E}[G^E_{\text{OPT}}] &\geq \theta_K \sum_{j=0}^{\ell-1} v_{t+j} + \theta_K b_{t+\ell} - \sum_{j=0}^{\ell} UCB_{b_{t+j},t} - UCB_{v_{t+\ell},t} \\
&\geq \theta_K \sum_{j=0}^{\ell-1} UCB_{v_{t+j},t} + \theta_K UCB_{b_{t+\ell},t} - \sum_{j=0}^{\ell} UCB_{b_{t+j},t} - UCB_{v_{t+\ell},t} + \\
&\quad - 2\theta_K \sum_{j=0}^{\ell-1} \beta_{v_{t+j},t} - 2\theta_K \beta_{b_{t+\ell},t}.
\end{aligned}
\tag{25}
$$

We show that (25) is lower bounded by $0$. From the $\text{ALG}^{\theta,U}$ perspective, the UCBs are equivalent to the actual weights. $\text{ALG}^{\theta,U}$ behaves exactly like $\text{ALG}^{\theta}$, and the UCBs satisfy the same constraints. Minimizing (25) under the UCB constraints is exactly equivalent to minimize (9) under the set of constraints defined in (10)-(13). Thus, substituting the UCBs with the actual values and following the same argument of the proof of Proposition 4, step (2a), we get that (25) is nonnegative for every sequence of UCBs.

Thus, in epoch $E$, we have

$$
\theta_K \mathbb{E}[G^E_{\text{ALG}^{\theta,U}}] - \mathbb{E}[G^E_{\text{OPT}}] \geq -2\theta_K \sum_{j=0}^{\ell-1} \beta_{v_{t+j},t} - 2\theta_K \beta_{b_{t+\ell},t}.
\tag{26}
$$

**(b)** $\{v_t, \ldots, v_{t+\ell}\}$**-type epochs.** Consider an epoch $E$ starting in $t$ and ending in $t+\ell$ of the second type. Then

$$
\mathbb{E}[G^E_{\text{OPT}}] \leq \sum_{j=0}^{\ell-1} UCB_{b_{t+j},t} + UCB_{v_{t+\ell},t},
$$

which is an overestimation of the total reward than any algorithm can achieve inside the epoch, due to epoch termination conditions of $\text{ALG}^{\theta,U}$. Note that under the good event all confidence intervals contain the true mean, for every packet type and time $t \in [T]$. We want to lower bound the following:

$$
\begin{aligned}
\theta_K \mathbb{E}[G^E_{\text{ALG}^{\theta}}] - \mathbb{E}[G^E_{\text{OPT}}] &\geq \theta_K \sum_{j=0}^{\ell} v_{t+j} - \sum_{j=0}^{\ell-1} UCB_{b_{t+j},t} - UCB_{v_{t+\ell},t} \\
&\geq \theta_K \sum_{j=0}^{\ell} UCB_{v_{t+j},t} - \sum_{j=0}^{\ell-1} UCB_{b_{t+j},t} - UCB_{v_{t+\ell},t} + \\
&\quad - 2\theta_K \sum_{j=0}^{\ell} \beta_{v_{t+j},t}.
\end{aligned}
\tag{27}
$$

We show that (27) is lower bounded by $0$. From the $\text{ALG}^{\theta,U}$ perspective, the UCBs are equivalent to the actual weights. $\text{ALG}^{\theta,U}$ behaves exactly like $\text{ALG}^{\theta}$, and the UCBs satisfy the same constraints. Minimizing (27) under the UCB constraints is exactly equivalent to minimize (18) under the set of constraints defined in (10)-(12) and (19). Thus, substituting the UCBs with the actual values and following the same argument of the proof of Proposition 4, step (2b), we get that (27) is nonnegative for every sequence of UCBs.

Thus, in epoch $E$, we have

$$
\theta_K \mathbb{E}[G^E_{\text{ALG}^{\theta,U}}] - \mathbb{E}[G^E_{\text{OPT}}] \geq -2\theta_K \sum_{j=0}^{\ell} \beta_{v_{t+j},t}.
\tag{28}
$$

**Bounding the Confidence Intervals.** We proved, through (26) and (28), that the per-epoch $\theta_K$-regret is upper bounded by the sum of the widths of the confidence intervals of the packet types selected during that epoch. Let $N_{i,T}$ be the number of times a packet of type $i \in [K]$ has been selected up to time $T$, and $K_E$ be the set of packet types selected during epoch $E$. Also, let $t_E$ the starting time of epoch $E$.

It is important to notice that, due to the constraints enforced on the UCBs of the selected packets, every packet type can only be selected at most once during an epoch. This is a consequence of the fact that the UCBs of the selected packets are strictly increasing during an epoch, and the UCBs are computed at the begin and kept fixed through $E$.

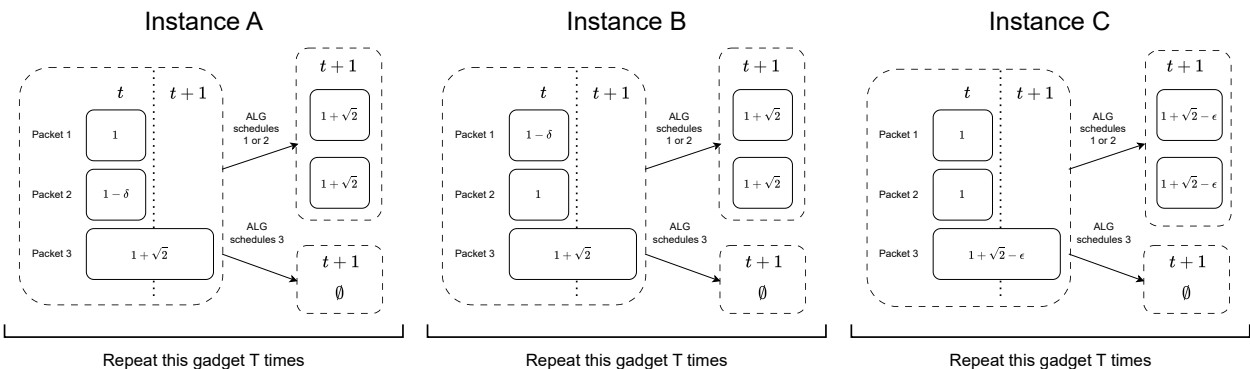

*Figure 1.* Exemplification of the lower bounds instances. Each gadget (of length two) is repeated $T$ times. What happens in the second round of a gadget depends on the action of ALG in the first.

To complete the proof, we sum over the epochs:

$$\mathbb{E}[G_{\text{OPT}}] - \theta_K \mathbb{E}[G_{\text{ALG}^{\theta,U}}] = \sum_E \mathbb{E}[G_{\text{OPT}}^E] - \theta_K \mathbb{E}[G_{\text{ALG}^{\theta,U}}^E]$$

$$\leq 2\theta_K \sum_E \sum_{i \in K_E} \beta_{i,t_E}$$

$$\leq 2\theta_K \sum_{i \in [K]} \sum_{j=1}^{N_{i,T}} \beta_{i,t_{i,j}}$$

$$\leq 2c\theta_K \sum_{i \in [K]} \sum_{j=1}^{N_{i,T}} \sqrt{\frac{\ln \frac{KT}{\delta}}{j}} \leq \widetilde{\mathcal{O}}\left(\sqrt{KT}\right),$$

where the $\widetilde{\mathcal{O}}$ only hides universal constants and logarithmic terms.

Setting $\delta = \frac{1}{T}$, the good event holds with probability at least $1 - \frac{1}{T}$ (Lemma A.1):

$$\mathbb{E}[R_{\theta_K,T}^{\text{ALG}^{\theta,U}}] = \mathbb{E}[R_{\theta_K,T}^{\text{ALG}^{\theta,U}} | \mathcal{E}(\delta)]\mathbb{P}\left(\mathcal{E}(\delta)\right) + \mathbb{E}[R_{\theta_K,T}^{\text{ALG}^{\theta,U}} | \mathcal{E}(\delta)^C]\mathbb{P}\left(\mathcal{E}(\delta)^C\right)$$

$$\leq \mathbb{E}[R_{\theta_K,T}^{\text{ALG}^{\theta,U}} | \mathcal{E}(\delta)] + T\frac{1}{T}$$

$$\leq \widetilde{\mathcal{O}}\left(\sqrt{KT}\right).$$

$\square$

**Theorem 4.3** (Lower Bound for $\theta_K$-regret). *In $2$-bounded $(K+1)$-OPSD instances, every ALG must satisfy*

$$\mathbb{E}[G_{OPT}] - \theta_K \mathbb{E}[G_{ALG}] \geq \Omega\left(\sqrt{T}\right).$$

*Proof.* Through the whole proof, we will refer to $\theta, x_1, \ldots, x_{K-1}$ as the solution to system 1. Note that $\theta < \Phi$ for every finite $K$. As the construction is very complex, for the sake of clarity, we start by proving the lower bound on a smaller problem for $\theta_K = \theta_2 = \sqrt{2}$. The full proof (generalized to an arbitrary $K$) can be found in the appendix.

**Lower Bound for $\theta_2$-regret.** We start by introducing some quantities. Note that $\theta_2 = \sqrt{2}$. Then, let $\delta = \frac{1}{\sqrt{T}}$, $\epsilon = \frac{1}{2\sqrt{T}}$, $\sigma = \frac{4}{\theta_2 - 1}$, $c = \frac{\theta_2 - 1}{8}$. Finally, let $X_0 \sim \mathcal{N}(1, \sigma)$ and $X_1 \sim \mathcal{N}(x_1, \sigma)$. Let ALG be an arbitrary, deterministic algorithm.

We consider three instances. Each instance is constructed by repeating a *gadget* for $T$ times. A gadget is a sub-problem that is self-contained; in our case, a gadget is composed of two subsequent rounds, starting from the first. Thus, the total number of rounds will be $2T$. In Figure 1

Instance A is constructed by repeating the following gadget $T$ times: in the first round $t$ of the gadget (*i.e.*, the odd rounds) we have $\mathcal{B}_t = \{(r_1 = t, d_1 = t, X_0, c_1 = 1), (r_2 = t, d_2 = t, X_0 - \delta, c_2 = 2), (r_3 = t, d_3 = t+1, X_1, c_3 = 3)\}$; in

the second round $t + 1$ of the gadget (*i.e.*, the even rounds) we have $\mathcal{B}_{t+1} = \varnothing$ if ALG sent the packet of type 3 in $t$, and $\mathcal{B}_{t+1} = \{(r_3 = t, d_3 = t + 1, X_1, c_3 = 3), (r_3 = t, d_3 = t + 1, X_1, c_3 = 3)\}$ if ALG sent a packet of type 1 or 2 in $t$. Intuitively, ALG is faced with two expiring packets (one having a slightly worse mean reward) and one packet that can also be sent in the following round. If ALG sends the latter, no more packets arrive, and the $\theta_2$-regret w.r.t. to OPT would be $r_2^A = 0$ (by the definition of $\theta_2$ and $x_1$). If ALG acts greedily and sends one of the former, then a copy of the third packet arrives and the expected $\theta_2$-regret w.t. to OPT would be $r_1^A = 0$ by scheduling the first and $\widetilde{r}_1^A = \delta\theta_K$ by scheduling the second.

Instance B is analogous to A, with the difference that we flip type 1 and type 2. As a consequence, $r_1^B = \widetilde{r}_1^A$ and $\widetilde{r}_1^B = 0$.

Instance C is constructed by repeating the following gadget $T$ times: in the first round $t$ of the gadget (*i.e.*, the odd rounds) we have $\mathcal{B}_t = \{(r_1 = t, d_1 = t, X_0, c_1 = 1), (r_2 = t, d_2 = t, X_0, c_2 = 2), (r_3 = t, d_3 = t + 1, X_1 - \epsilon, c_3 = 3)\}$; in the second round $t + 1$ of the gadget (*i.e.*, the even rounds) we have $\mathcal{B}_{t+1} = \varnothing$ if ALG sent the packet of type 3 in $t$, and $\mathcal{B}_{t+1} = \{(r_3 = t, d_3 = t + 1, X_1 - \epsilon, c_3 = 3), (r_3 = t, d_3 = t + 1, X_1 - \epsilon, c_3 = 3)\}$ if ALG sent a packet of type 1 or 2 in $t$. Intuitively, ALG is faced with two identical expiring packets (even if ALG doesn't know this) and one packet that can also be sent in the following round. If ALG sends the latter, no more packets arrive, and the $\theta_2$-regret w.r.t. to OPT would be $r_2^C = (\theta_2 - 1)\epsilon$ (by the definitions of $\theta_2$ and $x_1$). If ALG acts greedily and sends one of the former, then a copy of the third packet arrives and the expected $\theta_2$-regret w.r.t. to OPT would be $r_1^C = \widetilde{r}_1^C = (\theta_2 - 2)\epsilon < 0$.

Now we show that, no matter what the algorithm does, it is not possible to achieve $\mathcal{O}\left(\sqrt{T}\right)$ $\theta_2$-regret in A and C without achieving $\Omega\left(\sqrt{T}\right)$ $\theta_2$-regret in B. We call $N_{1,T}$, $\widetilde{N}_{1,T}$ and $N_{2,T}$ the number of times ALG selected packet type 1, 2, and 3 in the first step of a gadget, respectively. Let's write the expected regrets for every instance:

$$\mathbb{E}_A[R_{\theta_2,T}] = \theta_2\delta\mathbb{E}_A[\widetilde{N}_{1,T}],$$
$$\mathbb{E}_B[R_{\theta_2,T}] = \theta_2\delta\mathbb{E}_B[N_{1,T}],$$
$$\mathbb{E}_C[R_{\theta_2,T}] = (\theta_2 - 2)\epsilon\mathbb{E}_C[N_{1,T} + \widetilde{N}_{1,T}] + (\theta_2 - 1)\epsilon\mathbb{E}_C[N_{2,T}].$$

We force both $\mathbb{E}_A[R_{\theta_2,T}]$ and $\mathbb{E}_C[R_{\theta_2,T}]$ to be $\mathcal{O}\left(\sqrt{T}\right)$.

$$\mathbb{E}_A[R_{\theta_2,T}] = \theta_2\delta\mathbb{E}_A[\widetilde{N}_{1,T}] \leq c\sqrt{T},$$
$$\mathbb{E}_C[R_{\theta_2,T}] = (\theta_2 - 2)\epsilon\mathbb{E}_C[N_{1,T} + \widetilde{N}_{1,T}] + (\theta_2 - 1)\epsilon\mathbb{E}_C[N_{2,T}]$$
$$= \epsilon\mathbb{E}_C[N_{2,T}] + T(\theta_2 - 2)\epsilon \leq c\sqrt{T}.$$

These conditions yield an upper bound on the expected number of times that *bad* selections can be made in the respective instances. However, these selections are the *good* ones in Instance B.

As is common in the bandit literature, we now show that the instances are close from a statistical point of view, and thus hard to distinguish, and then use these constraints to force the expected regret in Instance B to be above a certain threshold. We want to upper bound the KL divergence between the instances. We use Lemma 1 from (Gerchinovitz & Lattimore, 2016), which allows us to decompose the KL between two instances after $T$ gadgets ($2T$ rounds). Note that, in every gadget, packet type $c_3$ is selected at most two times, while packets of types $c_1$ and $c_2$ can only be selected once and are mutually exclusive.

$$KL_T(\text{B},\text{A}) = \mathbb{E}_B[N_{1,T}]KL\left(\mathcal{N}(1 - \delta, \sigma), \mathcal{N}(1, \sigma)\right) + \mathbb{E}_B[\widetilde{N}_{1,T}]KL\left(\mathcal{N}(1, \sigma), \mathcal{N}(1 - \delta, \sigma)\right)$$
$$\leq 2T\frac{\delta^2}{\sigma^2},$$
$$KL_T(\text{B},\text{C}) = \mathbb{E}_B[N_{1,T}]KL\left(\mathcal{N}(1 - \delta, \sigma), \mathcal{N}(1, \sigma)\right) + 2\mathbb{E}_B[N_{2,T}]KL\left(\mathcal{N}(x_1, \sigma), \mathcal{N}(x_1 - \epsilon, \sigma)\right)$$
$$\leq T\frac{\delta^2 + 2\epsilon^2}{\sigma^2},$$

where the upper bound for the KL between two normal random variables is actually tight, since they all share the same variance $\sigma^2$.

Now, we bound the expected number of pulls in B with their corresponding numbers in A and C, respectively. We make use

of a classic Pinsker Inequality argument, plus the conditions previously derived.

$$\mathbb{E}_B[\widetilde{N}_{1,T}] \le \mathbb{E}_A[\widetilde{N}_{1,T}] + T\sqrt{\frac{1}{2}KL_T(\text{B},\text{A})}$$

$$\le \frac{c}{\theta_2\delta}\sqrt{T} + \sqrt{\frac{\delta^2}{\sigma^2}T^3},$$

$$\mathbb{E}_B[N_{2,T}] \le \mathbb{E}_C[N_{2,T}] + T\sqrt{\frac{1}{2}KL_T(\text{B},\text{C})}$$

$$\le \frac{c}{\epsilon}\sqrt{T} + (2-\theta_2)T + \sqrt{\frac{\delta^2 + 2\epsilon^2}{2\sigma^2}T^3}.$$

Finally, we wrap up by explicitly lower bounding the regret in Instance B.

$$\begin{aligned}
\mathbb{E}_B[R_{\theta_2,T}] &= \theta_2\delta\mathbb{E}_B[N_{1,T}] \\
&= \theta_2\delta(T - \mathbb{E}_B[\widetilde{N}_{1,T}] - \mathbb{E}_B[N_{2,T}]) \\
&\ge \theta_2\delta\left(T - \frac{c}{\theta_2\delta}\sqrt{T} + \sqrt{\frac{\delta^2}{\sigma^2}T^3} - \frac{c}{\epsilon}\sqrt{T} + (2-\theta_2)T + \sqrt{\frac{\delta^2 + 2\epsilon^2}{2\sigma^2}T^3}\right) \\
&= \frac{c}{4}\sqrt{T},
\end{aligned}$$

where the last inequality is obtained by simply plugging in the values defined at the beginning of the paragraph. To conclude, any algorithm ALG faced with Instances A, B, and C will have its regret lower bounded by $\frac{c}{4}\sqrt{T}$ in at least one of the three. Otherwise, we incur a contradiction.

$\square$

## B.2 Randomized Algorithms

**Theorem 5.1** (Upper Bound for the $\frac{5}{4}$-regret of $\text{ALG}^{R2}$). *In 2-bounded $K$-OPSD instances where randomization is allowed, $\text{ALG}^{R2}$ satisfies*

$$\mathbb{E}[G_{OPT}] \le \frac{5}{4}\mathbb{E}[G_{ALG^{R2}}] + \widetilde{\mathcal{O}}\left(\sqrt{KT}\right). \tag{2}$$

*Proof.* Assume that the good event $\mathcal{E}(\delta')$ holds, for some $\delta'$ to be specified later. We follow similar steps as Theorem 4.1 of (Bienkowski et al., 2011). Without loss of generality, we assume that exactly one packet with deadline $t$ and one packet with deadline $t + 1$ arrive at each round. All packets with deadline $t$ can be ignored except for the one with the largest upper confidence bound, and if no packets arrive, we can imagine a fictitious one with zero weight. At round $t$, all packets with deadline $t + 1$ except for the one with the largest UCB can be ignored, and all of the others can just be considered in round $t + 1$ as new packets.

We introduce a *potential function* $F(\cdot)$, a function that maps every state to a real (positive) number. This function, intuitively, quantifies the advantage of $\text{ALG}^{R2}$ w.r.t. OPT. When the state is empty, *i.e.*, both $\text{ALG}^{R2}$ and OPT buffers are empty, the potential is $F(\varnothing) = 0$. Thus, both at the start and at the end of the trial, the potential is null. Moreover, the potential is always nonnegative.

Consider round $t - 1$, the heaviest packet with deadline $t$ is $x_{t-1}^*$, and the packet with deadline $t$ and largest UCB is $x_{t-1}$. Let $\sigma_{t-1}$ be the probability with which $\text{ALG}^{R2}$ sends $x_{t-1}$. Let $z_{t-1}^* \in \{0, x_{t-1}^*\}$ be a dummy variable s.t. $z_{t-1}^* = 0$ if OPT sends $x_{t-1}^*$ during round $t - 1$ and $z_{t-1}^* = x_{t-1}^*$ otherwise (*i.e.*, $x_{t-1}^*$ is still in OPT buffer at round $t$). We call the triple $(x_{t-1}, \sigma_{t-1}, z_{t-1}^*)$ the *state* during round $t - 1$, and define the potential at round $t - 1$ as follows:

$$F(x_{t-1}, \sigma_{t-1}, z_{t-1}^*) = \begin{cases} 0, & z_{t-1}^* = 0 \\ \frac{1}{4}x_{t-1}\max\{5\sigma_{t-1} - 1, 3\sigma_{t-1}\}, & z_{t-1}^* = x_{t-1}^*. \end{cases} \tag{29}$$

Similarly, we describe the state during round $t$. Let $b_t$ be the packet with the largest UCB during round $t$, and $b_t^*$ be the heaviest. Similarly, consider $a_t$ and $a_t^*$. Let $v_t = a_t$ if $UCB_{a_t,t} \ge UCB_{x_{t-1},t}$, and $v_t = x_{t-1}$ otherwise. Let $\sigma_t$ be the

probability with which $\text{ALG}^{R2}$ sends $b_t$ at round $t$, then we observe that $\sigma_t = \sigma_{t-1}\widehat{q}_{a_t,b_t} + (1 - \sigma_{t-1})\widehat{q}_{v_t,b_t}$. Moreover, $z_t^* = 0$ if OPT sends $b_t^*$ during $t$ and $z_t^* = b_t^*$ otherwise. The potential during round $t$ is then $F(b_t, \sigma_t, z_t^*)$.

To prove our statement, we first prove that for every round $t \in [T]$ the following holds:

$$1.25\mathbb{E}[\Delta G^t_{\text{ALG}^{R2}}] + F(x_{t-1}, \sigma_{t-1}, z_{t-1}^*) - F(b_t, \sigma_t, z_t^*) \geq$$
$$\geq \mathbb{E}[\Delta G^t_{\text{OPT}}] - c(\sigma_{t-1}\beta_{a_t,t} + \sigma_{t-1}\beta_{x_{t-1},t} + (1 - \sigma_{t-1})\beta_{v_t,t} + \beta_{b_t,t}), \tag{30}$$

where $\Delta G^t_{\text{ALG}^{R2}}$ and $\Delta G^t_{\text{OPT}}$ are the expected weights of the packets sent at time $t$ by $\text{ALG}^{R2}$ and OPT, respectively. We prove Equation (30) under every possible scenario. For this first section of the proof, we assume without loss of generality that the burn-in period has finished. The regret accrued during that time represents an $\mathcal{O}(\log T)$ additive factor which is dominated by $\mathcal{O}(\sqrt{T})$.

**Case 1: OPT sends $b_t^*$** In this case, $\mathbb{E}[\Delta G^t_{\text{OPT}}] = b_t^*$ and $z_t^* = 0$. Thus, $F(b_t, \sigma_t, z_t^*) = 0$. As the potential is always nonnegative, we are left to prove that $1.25\mathbb{E}[\Delta G^t_{\text{ALG}^{R2}}] \geq b_t^* - c(\sigma_{t-1}\beta_{a_t,t} + (1 - \sigma_{t-1}\beta_{v_t,t} + \beta_{b_t,t})$. This can be proven by observing that $\text{ALG}^{R2}$ selects $a_t$ with probability $\sigma_{t-1}\widehat{p}_{a_t b_t,t}$, $a_t$ with probability $(1 - \sigma_{t-1})\widehat{p}_{v_t b_t,t}$, and $b_t$ with probability $\sigma_t$. Thus

$$\frac{5}{4}\mathbb{E}[\Delta G^t_{\text{ALG}^{R2}}] = \frac{5}{4}\sigma_{t-1}(\widehat{p}_{a_t,b_t,t}v_t + \widehat{q}_{a_t,b_t}b_t) + \frac{5}{4}(1 - \sigma_{t-1})(\widehat{p}_{v_t,b_t,t}v_t + \widehat{q}_{v_t,b_t}b_t)$$
$$\overset{(6b)}{\geq} \sigma_{t-1}\left(b_t^* - \frac{5}{3}\beta_{a_t,t} - 2\beta_{b_t,t}\right) + (1 - \sigma_{t-1})\left(b_t^* - \frac{5}{3}\beta_{v_t,t} - 2\beta_{b_t,t}\right)$$
$$= b_t^* - \frac{5}{3}\sigma_{t-1}\beta_{a_t,t} - \frac{5}{3}(1 - \sigma_{t-1})\beta_{v_t,t} - 2\beta_{b_t,t}.$$

**Case 2: OPT does not send $b_t^*$** In this case, $z_t^* = b_t^*$. Thus, $F(b_t, \sigma_t, z_t^*) = \frac{1}{4}b_t \max\{5\sigma_t - 1, 3\sigma_{t-1}\}$. We have to prove that

$$\frac{5}{4}\mathbb{E}[\Delta G^t_{\text{ALG}^{R2}}] = \frac{1}{4}\min\{b_t, 2\sigma_t b_t\} + \frac{5}{4}\sigma_{t-1}\widehat{p}_{a_t b_t,t}a_t + \frac{5}{4}(1 - \sigma_t)\widehat{p}_{v_t b_t,t}v_t + F(x_{t-1}, \sigma_{t-1}, z_{t-1}^*)$$
$$\geq \mathbb{E}[\Delta G^t_{\text{OPT}}] - c(\beta_{a_t,t} + \beta_{v_t,t} + \beta_{b_t,t}).$$

**Case $2a$: OPT sends $a_t^*$** Let $v_t^* \coloneqq \max\{a_t^*, x_{t-1}^*\}$. In this case, $\mathbb{E}[\Delta G^t_{\text{OPT}}] = a_t^* \leq v_t^*$. Moreover, $F(x_{t-1}, \sigma_{t-1}, z_{t-1}^*) \geq 0$. We first consider the case $b_t \leq 2\sigma_t b_t$:

$$\frac{1}{4}b_t + \frac{5}{4}\sigma_{t-1}\widehat{p}_{a_t b_t,t}a_t + \frac{5}{4}(1 - \sigma_{t-q})\widehat{p}_{v_t b_t,t}v_t \overset{(6a)}{\geq} \sigma_{t-1}a_t^* + (1 - \sigma_{t-1})v_t^* - \frac{5}{3}\sigma_{t-1}\beta_{a_t,t} - \frac{5}{3}(1 - \sigma_{t-1})\beta_{v_t,t}$$
$$\geq a_t^* - \frac{5}{3}\sigma_{t-1}\beta_{a_t,t} - \frac{5}{3}(1 - \sigma_{t-1})\beta_{v_t,t}.$$

We then consider the case $b_t > 2\sigma_t b_t$:

$$\frac{1}{4}2\sigma_t b_t + \frac{5}{4}\sigma_{t-1}\widehat{p}_{a_t b_t,t}a_t + \frac{5}{4}(1 - \sigma_{t-q})\widehat{p}_{v_t b_t,t}v_t \overset{(6c)}{\geq} \sigma_{t-1}a_t^* + (1 - \sigma_{t-1})v_t^* - \frac{5}{3}\sigma_{t-1}\beta_{a_t,t} - \frac{5}{3}(1 - \sigma_{t-1})\beta_{v_t,t}$$
$$\geq a_t^* - \frac{5}{3}\sigma_{t-1}\beta_{a_t,t} - \frac{5}{3}(1 - \sigma_{t-1})\beta_{v_t,t}.$$

**Case** $2a$**: OPT sends** $x_{t-1}^*$    In this case, $\mathbb{E}[\Delta G_{\text{OPT}}^t] = x_{t-1}^* = v_t^* \geq x_{t-1}$. We first consider the case $UCB_{b_t,t} \leq LCB_{v_t,t}$.
Note that $F(x_{t-1}, \sigma_{t-1}, z_{t-1}^*) = F(x_{t-1}, \sigma_{t-1}, x_{t-1}^*) \geq \frac{1}{4}(5\sigma_{t-1} - 1)x_{t-1}$ and $\widehat{p}_{v_t b_t,t} = 1$, which implies

$$
\begin{aligned}
\frac{5}{4}\mathbb{E}[\Delta G_{\text{ALG}}^t] + F(x_{t-1}, \sigma_{t-1}, z_{t-1}^*) &\geq \frac{5}{4}(1 - \sigma_{t-1})\widehat{p}_{v_t b_t,t} v_t + F(x_{t-1}, \sigma_{t-1}, z_{t-1}^*) \\
&\geq \frac{5}{4}(1 - \sigma_{t-1})v_t^* - \frac{5}{4}(1 - \sigma_{t-1})\beta_{v_t,t} + \frac{1}{4}(5\sigma_{t-1} - 1)x_{t-1} \\
&\geq \left(\frac{5}{4}v_t^* - \frac{1}{4}x_{t-1}\right) + \frac{5}{4}\sigma_{t-1}(x_{t-1} - v_t^*) - \frac{5}{4}(1 - \sigma_{t-1})\beta_{v_t,t} \\
&\geq x_{t-1}^* + \frac{5}{4}\sigma_{t-1}(x_{t-1} - v_t^*) - \frac{5}{4}(1 - \sigma_{t-1})\beta_{v_t,t} \\
&= x_{t-1}^* + \frac{5}{4}\sigma_{t-1}(x_{t-1} - x_{t-1}^*) - \frac{5}{4}(1 - \sigma_{t-1})\beta_{v_t,t} \\
&\geq x_{t-1}^* - \frac{5}{4}\sigma_{t-1}\beta_{x_{t-1},t} - \frac{5}{4}(1 - \sigma_{t-1})\beta_{v_t,t}.
\end{aligned}
$$

Second, consider the scenario $UCB_{b_t,t} > LCB_{v_t,t}$ and $LCB_{b_t,t} < UCB_{v_t,t}$, then $\widehat{p}_{v_t b_t,t} = \frac{4}{5}$:

$$
\begin{aligned}
\frac{5}{4}\mathbb{E}[\Delta G_{\text{ALG}}^t] + F(x_{t-1}, \sigma_{t-1}, z_{t-1}^*) &\geq \frac{5}{4}(1 - \sigma_{t-1})\widehat{p}_{v_t b_t,t} v_t + \frac{5}{4}\sigma_t b_t + F(x_{t-1}, \sigma_{t-1}, z_{t-1}^*) \\
&\geq (1 - \sigma_{t-1})v_t^* - (1 - \sigma_{t-1})\beta_{v_t,t} + \frac{5}{4}\sigma_t b_t + F(x_{t-1}, \sigma_{t-1}, z_{t-1}^*) \\
&\geq \frac{1}{4}(4v_t^* + b_t - x_{t-1}) + \frac{5}{4}\sigma_{t-1}\left(x_{t-1} - \frac{4}{5}v_t^* - \frac{1}{5}b_t\right) - (1 - \sigma_{t-1})\beta_{v_t,t} \\
&\geq \frac{1}{4}(5v_t^* - x_{t-1}) + \frac{5}{4}\sigma_{t-1}(x_{t-1} - v_t^*) - (1 - \sigma_{t-1})\beta_{v_t,t} - \frac{1}{4}(1 + \sigma_{t-1})\beta_{b_t,t} \\
&\geq x_{t-1}^* - \frac{5}{4}\sigma_{t-1}\beta_{x_{t-1},t} - (1 - \sigma_{t-1})\beta_{v_t,t} - \frac{1}{2}\beta_{b_t,t}.
\end{aligned}
$$

Finally, consider the scenario $LCB_{b_t,t} > UCB_{v_t,t}$. If $b_t \leq 2\sigma_t b_t$ we have:

$$
\begin{aligned}
\frac{5}{4}\mathbb{E}[\Delta G_{\text{ALG}}^t] &\geq \frac{1}{4}b_t + \frac{5}{4}(1 - \sigma_{t-1})\widehat{p}_{v_t b_t,t} v_t \\
&\overset{(6a)}{\geq} \frac{1}{4}b_t^* + (1 - \sigma_{t-1})v_t^* - \frac{5}{3}(1 - \sigma_{t-1})\beta_{v_t,t}.
\end{aligned}
$$

If $b_t > 2\sigma_t b_t$ we have:

$$
\begin{aligned}
\frac{5}{4}\mathbb{E}[\Delta G_{\text{ALG}}^t] &\geq \frac{1}{4}2\sigma_t b_t + \frac{5}{4}\sigma_{t-1}\widehat{p}_{a_t b_t,t} a_t + \frac{5}{4}(1 - \sigma_{t-q})\widehat{p}_{v_t b_t,t} v_t \\
&= \frac{1}{4}\sigma_{t-1}(5\widehat{p}_{a_t b_t,t} a_t + 2\widehat{q}_{a_t b_t,t} b_t) + \frac{1}{4}(1 - \sigma_{t-1})(5\widehat{p}_{v_t b_t,t} v_t + 2\widehat{q}_{v_t b_t,t} b_t) \\
&\overset{(6d)}{\geq} \frac{1}{4}b_t^* + (1 - \sigma_{t-1})v_t^* - \frac{5}{3}\sigma_{t-1}\beta_{a_t,t} - \frac{5}{3}(1 - \sigma_{t-1})\beta_{v_t,t}.
\end{aligned}
$$

Noting that $F(x_{t-1}, \sigma_{t-1}, z_{t-1}^*) \geq \frac{3}{4}\sigma_{t-1}x_{t-1}$, we have:

$$
\begin{aligned}
\frac{5}{4}\mathbb{E}[\Delta G_{\text{ALG}}^t] + F(x_{t-1}, \sigma_{t-1}, z_{t-1}^*) &\geq \frac{1}{4}b_t^* + (1 - \sigma_{t-1})v_t^* + \frac{3}{4}\sigma_{t-1}x_{t-1} - \frac{5}{3}\sigma_{t-1}\beta_{a_t,t} - \frac{5}{3}(1 - \sigma_{t-1})\beta_{v_t,t} \\
&\geq v_t^* + \left(\frac{1}{4}b_t^* - \frac{1}{4}\sigma_{t-1}x_{t-1}\right) - \frac{5}{3}\sigma_{t-1}\beta_{a_t,t} - \frac{5}{3}(1 - \sigma_{t-1})\beta_{v_t,t} \\
&\geq v_t^* + \left(\frac{1}{4}b_t - \frac{1}{4}\sigma_{t-1}x_{t-1}\right) - \frac{5}{3}\sigma_{t-1}\beta_{a_t,t} - \frac{5}{3}(1 - \sigma_{t-1})\beta_{v_t,t} \\
&\geq v_t^* - \frac{5}{3}\sigma_{t-1}\beta_{a_t,t} - \frac{5}{3}(1 - \sigma_{t-1})\beta_{v_t,t}.
\end{aligned}
$$

**Bounding the Confidence Intervals**   We have proven that, whatever OPT does, it holds

$$1.25\mathbb{E}[\Delta G^t_{\text{ALG}^{R2}}] + F(x_{t-1}, \sigma_{t-1}, z^*_{t-1}) - F(b_t, \sigma_t, z^*_t) \geq$$
$$\geq \mathbb{E}[\Delta G^t_{\text{OPT}}] - 2\left(\sigma_{t-1}\beta_{a_t,t} - \sigma_{t-1}\beta_{x_{t-1},t} - (1 - \sigma_{t-1})\beta_{v_t,t} - \beta_{b_t,t}\mathbb{1}_{\{UCB_{b_t,t} > LCB_{v_t,t}\}}\right).$$

We start by lower bounding (in high probability) the number of times every packet is sent up to time $t$. Let $\mathcal{F}_{t-1}$ be the filtration up to time $t - 1$, *i.e.*, the $\sigma$-algebra generated by the sequence of events observed by $\text{ALG}^{R2}$. We define an auxiliary random variable $B_{i,t}$ which is conditionally Bernoulli s.t. $\mathbb{P}(B_{i,t} = 1 \mid \mathcal{F}_{t-1})$ represents the probability that packet $i$ is sent at time $t$. Let $N_{i,t}$ be the (stochastic) number of times that packet $i \in [K]$ was sent up to time $t \in [T]$, *i.e.* $N_{i,t} = \sum_{\ell=1}^t B_{i,\ell}$.

**(i) Bounding $\sum_{t=1}^T \sigma_{t-1}\beta_{a_t,t}$**   At time $t$, we have

$$\mathbb{P}(B_{a_t,t} = 1 \mid \mathcal{F}_{t-1}) \geq \sigma_{t-1}\widehat{p}_{a_t b_t, t} \geq \sigma_{t-1}\frac{1}{5}.$$

Fix $i \in [K]$. Let $G^a_{i,\ell} = \mathbb{1}_{\{a_l = i\}}$, and $E^a_{i,t} = \{\ell \in [t] : G^a_{i,\ell} = 1\}$. $G^a_{i,\ell}$ is $\mathcal{F}_{t-1}$-predictable for every $\ell \in [t]$. Let $N^a_{i,t} = \sum_{\ell=1}^t B_{a_\ell,\ell}G^a_{i,\ell}$. By algorithm's definition, for every $i \in [K]$, we get that $B_{i,t'}G^a_{i,t'} = 1$ for every $t'$ s.t. $\sum_{\ell=1}^{t'} G^a_{i,\ell} < 2\log\frac{1}{\delta}$. Thus, $N^a_{i,t'} \geq 2\log\frac{1}{\delta}$. Then, noting that $N^a_{i,t}\mathbb{P}(B_{i,t} = 1 \mid \mathcal{F}_{t-1})G^a_{i,t}$ is a martingale difference sequence, we let $\delta \in (0, 1)$ and $t \geq t'$, and by Azuma-Hoeffding Inequality with optional skipping we get

$$N^a_{i,t} = \sum_{\ell=1}^t B_{a_\ell,\ell}G^a_{i,\ell}$$

$$\geq \sum_{\ell=1}^t \sigma_{\ell-1}\frac{1}{5}G^a_{i,\ell} - \sqrt{\frac{1}{10}\sum_{\ell=1}^t G^a_{i,\ell}\log\frac{1}{\delta}}$$

$$\geq N^a_{i,t'} + \frac{1}{10}\sum_{\ell=t'}^t \sigma_{\ell-1}G^a_{i,\ell-1}$$

$$\geq 2\log\frac{1}{\delta} + \frac{1}{10}\sum_{\ell=t'}^t \sigma_{\ell-1}G^a_{i,\ell-1}.$$

It follows that, for every $i \in [K]$:

$$\sum_{t=1}^T \frac{\sigma_{t-1}}{\sqrt{N_{i,t}}}G^a_{i,t} \leq t' + \sum_{t \in E^a_{i,t} \cap \{t \geq t'\}} \frac{\sigma_{t-1}}{\sqrt{N^a_{i,t}}}$$

$$\leq 2\log\frac{1}{\delta} + \sqrt{25E^a_{i,t}}.$$

Thus, since $\beta_{a_{t-1},t} = c\sqrt{\frac{\ln\frac{KT}{\delta'}}{N^a_{i,t}}}$, for some $\delta' \in (0, 1)$, we have that the following holds

$$\sum_{t=1}^T \sigma_{t-1}\beta_{a_{t-1},t} \leq \sum_{i \in [K]} \sum_{t \in E^a_{i,T}} c\sigma_{t-1}\sqrt{\frac{\ln\frac{KT}{\delta'}}{N^a_{i,t}}}$$

$$\leq 2cK\log\frac{1}{\delta}\sqrt{\ln\frac{KT}{\delta'}} + c\sum_{i \in [K]}\sqrt{25E^a_{i,t}\ln\frac{KT}{\delta'}}$$

$$\leq 2cK\log\frac{1}{\delta}\sqrt{\ln\frac{KT}{\delta'}} + c\sqrt{25KT\ln\frac{KT}{\delta'}},$$

where the last inequality follows from the concavity of the square root.

**(ii) Bounding $\sum_{t=1}^T \beta_{b_t,t}\mathbb{1}_{\{UCB_{b_t,t} > LCB_{v_t,t}\}}$**   At time $t$, we have

$$\mathbb{P}(B_{b_t,t} = 1 \mid \mathcal{F}_{t-1}) \geq \sigma_t \geq \frac{1}{5}\mathbb{1}_{\{UCB_{b_t,t} > LCB_{v_t,t}\}}$$

Let $G_{i,\ell}^b = \mathbb{1}_{\{b_l=i\}\cap\{UCB_{b_t,t}>LCB_{v_t,t}\}}$ and $E_{i,t}^b = \{\ell \in [t] : G_\ell^b = 1\}$. $G_{i,\ell}^b$ is $\mathcal{F}_{t-1}$-predictable for every $\ell \in [t]$. Note that $G_t^b(B_{i,t} - \mathbb{P}(B_{i,t} = 1 \mid \mathcal{F}_{t-1}))$ is a martingale difference sequence. By algorithm's definition, for every $i \in [K]$, we get that $B_{i,t'}G_{i,t'}^b = 1$ for every $t'$ s.t. $\sum_{\ell=1}^{t'} G_{i,\ell}^b < 50 \log \frac{1}{\delta}$. Thus, $N_{i,t'}^b \geq 50 \log \frac{1}{\delta}$. Let $\delta \in (0,1)$ and $t \geq t'$, then we use the Azuma-Hoeffding inequality with optional skipping to get that, with probability at least $1 - \delta$, it holds

$$N_{i,t}^b \geq \sum_{\ell=1}^t G_{i,\ell}^b \mathbb{P}(B_{i,\ell} = 1 \mid \mathcal{F}_{\ell-1}) - \sqrt{\frac{1}{2} \sum_{\ell=1}^t G_{i,\ell}^b \mathbb{P}(B_{i,\ell} = 1 \mid \mathcal{F}_{\ell-1}) \log \frac{1}{\delta}}$$

$$\geq \frac{1}{5} \sum_{\ell=1}^t G_{i,\ell}^b - \sqrt{\frac{1}{2} \sum_{\ell=1}^t G_{i,\ell}^b \log \frac{1}{\delta}}$$

$$\geq \frac{1}{5} \sum_{\ell=1}^t G_{i,\ell}^b - \sqrt{\frac{1}{2} \sum_{\ell=1}^t G_{i,\ell}^b \log \frac{1}{\delta}}$$

$$\geq 50 \log \frac{1}{\delta} + \frac{1}{10} \sum_{\ell=t'}^t G_{i,\ell}^b.$$

It follows that, for every $i \in [K]$:

$$\sum_{t=1}^T \frac{1}{\sqrt{N_{i,t}}} G_{i,t}^b \leq t' + \sum_{t \in E_{i,t}^b \cap \{t \geq t'\}} \frac{1}{\sqrt{N_{i,t}^b}}$$

$$\leq 50 \log \frac{1}{\delta} + \sqrt{2E_{i,t}^b}.$$

Thus, since $\beta_{b_t,t} \leq c\sqrt{\frac{\ln \frac{KT}{\delta'}}{N_{i,t}^b}}$, for some $\delta' \in (0,1)$, we have that the following holds

$$\sum_{t=1}^T \beta_{b_t,t} \leq \sum_{i \in [K]} \sum_{t \in E_{i,T}^b} c\sigma_{t-1}\sqrt{\frac{\ln \frac{KT}{\delta'}}{N_{i,t}^b}}$$

$$\leq c50K \log \frac{1}{\delta} \sqrt{\ln \frac{KT}{\delta'}} + c \sum_{i \in [K]} \sqrt{50E_{i,t}^b \ln \frac{KT}{\delta'}}$$

$$\leq c50K \log \frac{1}{\delta} \sqrt{\ln \frac{KT}{\delta'}} + c\sqrt{50KT \ln \frac{KT}{\delta'}}.$$

where the last inequality follows from the concavity of the square root.

**(iii) Bounding $\sum_{t=1}^T (1 - \sigma_{t-1})\beta_{v_t,t}$** At time $t$, we have

$$\mathbb{P}(B_{v_t,t} = 1 \mid \mathcal{F}_{t-1}) \geq (1 - \sigma_{t-1})\widehat{p}_{v_t b_t,t} \geq (1 - \sigma_{t-1})\frac{1}{5}.$$

Following an analogous procedure as in paragraph (i), we get

$$\sum_{t=1}^T (1 - \sigma_{t-1})\beta_{v_t,t} \leq \sum_{i \in [K]} \sum_{t \in E_{i,T}^v} c(1 - \sigma_{t-1})\sqrt{\frac{\ln \frac{KT}{\delta'}}{N_{i,t}^v}}$$

$$\leq 2cK \log \frac{1}{\delta} \sqrt{\ln \frac{KT}{\delta'}} + c \sum_{i \in [K]} \sqrt{25E_{i,t}^v \ln \frac{KT}{\delta'}}$$

$$\leq 2cK \log \frac{1}{\delta} \sqrt{\ln \frac{KT}{\delta'}} + c\sqrt{25KT \ln \frac{KT}{\delta'}}.$$

**(iv) Bounding $\sum_{t=1}^T \sigma_{t-1}\beta_{x_{t-1},t}$** Fix $i \in [K]$. Let $G_{i,\ell-1}^x = \mathbb{1}_{\{x_{l-1}=i\}}$, and $E_{i,t}^x = \{\ell \in [t] : G_{i,\ell-1}^x = 1\}$. $G_{i,\ell-1}^x$ is $\mathcal{F}_{t-2}$-predictable for every $\ell \in [t]$. Also, $B_{x_{t-1},t-1}$ belongs to $\mathcal{F}_{t-1}$. Note that $\mathbb{P}(B_{x_{t-1},t-1} = 1 \mid \mathcal{F}_{t-2}) = \sigma_{t-1}$. Let

$N_{i,t-1}^x = \sum_{\ell=1}^{t} B_{x_{\ell-1},\ell-1} G_{i,\ell-1}^x$. By algorithm's definition, for every $i \in [K]$, we get that $B_{i,t'} G_{i,t'}^x = 1$ for every $t'$ s.t. $\sum_{\ell=1}^{t'} G_{i,\ell}^x < 2\log\frac{1}{\delta}$. Thus, $N_{i,t'}^x \geq 2\log\frac{1}{\delta}$. Then, noting that $N_{i,t-1}^x - \sum_{\ell=1}^{t} \sigma_{\ell-1} G_{i,\ell-1}^x$ is a martingale difference sequence, we let $\delta \in (0,1)$ and $t \geq t'$, and by Azuma-Hoeffding Inequality with optional skipping we get

$$
\begin{aligned}
N_{i,t-1}^x &= \sum_{\ell=1}^{t} B_{x_{\ell-1},\ell-1} G_{i,\ell-1}^x \\
&\geq \sum_{\ell=1}^{t} \sigma_{\ell-1} G_{i,\ell-1}^x - \sqrt{\frac{1}{2} \sum_{\ell=1}^{t} \sigma_{\ell-1} G_{i,\ell-1}^x \log\frac{1}{\delta}} \\
&\geq N_{i,t'}^x + \frac{1}{2} \sum_{\ell=t'}^{t} \sigma_{\ell-1} G_{i,\ell-1}^x \\
&\geq 2\log\frac{1}{\delta} + \frac{1}{2} \sum_{\ell=t'}^{t} \sigma_{\ell-1} G_{i,\ell-1}^x.
\end{aligned}
$$

It follows that, for every $i \in [K]$:

$$
\begin{aligned}
\sum_{t=1}^{T} \frac{\sigma_{t-1}}{\sqrt{N_{i,t}}} G_{i,t-1}^x &\leq t' + \sum_{t \in E_{i,t}^x \cap \{t \geq t'\}} \frac{\sigma_{t-1}}{\sqrt{N_{i,t}^x}} \\
&\leq 2\log\frac{1}{\delta} + \sqrt{2 E_{i,t}^x}.
\end{aligned}
$$

Thus, since $\beta_{x_{t-1},t} \leq c\sqrt{\frac{\ln\frac{KT}{\delta'}}{N_{i,t}^x}}$, for some $\delta' \in (0,1)$, we have that the following holds

$$
\begin{aligned}
\sum_{t=1}^{T} \sigma_{t-1} \beta_{x_{t-1},t} &\leq \sum_{i \in [K]} \sum_{t \in E_{i,T}^x} c\sigma_{t-1} \sqrt{\frac{\ln\frac{KT}{\delta'}}{N_{i,t}^x}} \\
&\leq 2cK\log\frac{1}{\delta} \sqrt{\ln\frac{KT}{\delta'}} + c \sum_{i \in [K]} \sqrt{2 E_{i,t}^x \ln\frac{KT}{\delta'}} \\
&\leq 2cK\log\frac{1}{\delta} \sqrt{\ln\frac{KT}{\delta'}} + c\sqrt{2KT \ln\frac{KT}{\delta'}},
\end{aligned}
$$

where the last inequality follows from the concavity of the square root.

Setting $\delta' = \frac{1}{T}$ we get that $\mathbb{P}(\mathcal{E}(\delta')) \geq 1 - \frac{1}{T}$ (Lemma A.1). Setting $\delta = \frac{1}{KT^2}$ (to account for the $K$ bounds to hold simultaneously at every round), and by a standard union bound argument, we get that all Azuma-Hoeffding bounds used in paragraphs (i)-(iv) hold simultaneously with probability greater than $1 - \frac{1}{KT^2}$. We call this event $\widetilde{\mathcal{E}}(\delta)$ and we get $\mathbb{P}(\widetilde{\mathcal{E}}(\delta')) \geq 1 - \frac{1}{KT^2}$.

$$
\begin{aligned}
\mathbb{E}[R_{\frac{5}{4},T}^{\mathrm{ALG}^{R2}}] &= \mathbb{E}[R_{\frac{5}{4},T}^{\mathrm{ALG}^{R2}} | \mathcal{E}(\delta) \cap \widetilde{\mathcal{E}}(\delta)] \mathbb{P}(\mathcal{E}(\delta) \cap \widetilde{\mathcal{E}}(\delta)) + \\
&\quad + \mathbb{E}[R_{\frac{5}{4},T}^{\mathrm{ALG}^{R2}} | (\mathcal{E}(\delta) \cap \widetilde{\mathcal{E}}(\delta))^C] \mathbb{P}((\mathcal{E}(\delta) \cap \widetilde{\mathcal{E}}(\delta))^C) \\
&\leq \widetilde{\mathcal{O}}(\sqrt{KT}).
\end{aligned}
$$

Putting all together completes the proof. $\qquad\square$

**Theorem 5.2** (Upper Bound for the $\frac{e}{e-1}$-regret of $\mathrm{ALG}^{Rs}$). *In s-bounded K-OPSD instances where randomization is allowed, $\mathrm{ALG}^{Rs}$ satisfies*

$$
\mathbb{E}[G_{OPT}] \leq \frac{e}{e-1} \mathbb{E}[G_{\mathrm{ALG}^{Rs}}] + \widetilde{\mathcal{O}}(\sqrt{KT}), \tag{4}
$$

*for every $s > 1$.*

*Proof.* Assume that the good event $\mathcal{E}(\delta)$ holds, for some $\delta$ to be specified later. Without loss of generality, assume that $\mathrm{OPT}$ schedules its packets in an increasing order of deadline. We define the set $\mathcal{B}_t^{\mathrm{OPT}}$ as the set of packets in the buffer of $\mathrm{OPT}$ at round $t$ that $\mathrm{OPT}$ will schedule in the future rounds. We introduce a potential function $F(\cdot)$ defined as

$$F(\mathcal{B}_t, \mathcal{B}_t^{\mathrm{OPT}}) = \sum_{w \in \mathcal{B}_t^{\mathrm{OPT}} \setminus \mathcal{B}_t} w.$$

Incoming new packets and expirations do not modify $F$, and $F$ only varies when $\mathrm{ALG}^{Rs}$ executes a packet. Let $f_t$ be the packet selected at time $t$ by $\mathrm{ALG}^{Rs}$, and $j_t$ be the packet selected at time $t$ by $\mathrm{OPT}$.

**Case 1:** $j_t \in \mathcal{B}_t^{\mathbf{OPT}} \setminus \mathcal{B}_t$     In this case, $F$ decreases by $j_t$, and increases by $f_t$, thus $\Delta_t F = f_t - j_t$. Thus $j_t - \frac{e}{e-1} f_t + \Delta_t F = -\frac{1}{e-1} f_t \leq 0$.

**Case 2:** $j_t \in \mathcal{B}_t^{\mathbf{OPT}} \cap \mathcal{B}_t$     In this case, $F$ increases at most by $f_t$, thus $\Delta_t F \leq f_t$.

If $UCB_{j_t,t} \geq e^{x_t} LCB_{\underline{h}_t,t}$, then $\Delta_t F = 0$. Indeed, if $j_t$ satisfies the $\mathrm{ALG}^{Rs}$ condition, it is either $j_t = f_t$ or $f_t$ has a closer deadline than $j_t$. As $\mathrm{OPT}$ breaks ties in favor of the earliest deadline, then $f_t \notin \mathcal{B}_{t+1}^{\mathrm{OPT}}$, and $\Delta_t F = 0$.

Let $z_t = \max\left\{ -1 + \ln \frac{UCB_{\bar{h}_t,t}}{LCB_{\underline{h}_t,t}}, \ln \frac{UCB_{j_t,t}}{LCB_{\underline{h}_t,t}} \right\}$. Then, if $x_t \in \left[ -1 + \ln \frac{UCB_{\bar{h}_t,t}}{LCB_{\underline{h}_t,t}}, z_t \right]$ we have $\Delta_t F = 0$. Else, we have $0 \leq \Delta_t F \leq f_t$. We can then take the expectation also w.r.t. to the randomization of $x_t$ and write:

$$\mathbb{E}_{x_t}\left[ j_t - \frac{e}{e-1} f_t + \Delta_t F \right] = j_t - \frac{1}{e-1} \mathbb{E}_{x_t}[f_t] - \mathbb{E}_{x_t}[f_t - \Delta_t F]$$

$$\leq j_t - \frac{1}{e-1} \mathbb{E}_{x_t}[UCB_{f_t,t} - \beta_{f_t,t}] - \mathbb{E}_{x_t}[UCB_{f_t,t} - \beta_{f_t,t} - \Delta_t F]$$

$$\leq j_t - \frac{1}{e-1} \int_{-1+\ln \frac{UCB_{\bar{h}_t,t}}{LCB_{\underline{h}_t,t}}}^{\ln \frac{UCB_{\bar{h}_t,t}}{LCB_{\underline{h}_t,t}}} e^x LCB_{\underline{h}_t,t} dx - \int_{-1+\ln \frac{UCB_{\bar{h}_t,t}}{LCB_{\underline{h}_t,t}}}^{z} e^x LCB_{\underline{h}_t,t} dx + 2\mathbb{E}_{x_t}[\beta_{f_t,t}]$$

$$\leq j_t - \frac{1}{e-1}\left( \frac{UCB_{\bar{h}_t,t}}{LCB_{\underline{h}_t,t}} - \frac{1}{e}\frac{UCB_{\bar{h}_t,t}}{LCB_{\underline{h}_t,t}} \right) LCB_{\underline{h}_t,t} - \left( \frac{UCB_{j_t,t}}{LCB_{\underline{h}_t,t}} - \frac{1}{e}\frac{UCB_{\bar{h}_t,t}}{LCB_{\underline{h}_t,t}} \right) LCB_{\underline{h}_t,t} + 2\mathbb{E}_{x_t}[\beta_{f_t,t}]$$

$$\leq j_t - UCB_{j_t,t} - LCB_{\underline{h}_t,t}\left( \frac{1}{e-1}\frac{UCB_{\bar{h}_t,t}}{LCB_{\underline{h}_t,t}} - \frac{1}{e-1}\frac{UCB_{\bar{h}_t,t}}{LCB_{\underline{h}_t,t}} \right) + 2\mathbb{E}_{x_t}[\beta_{f_t,t}]$$

$$\leq 2\mathbb{E}_{x_t}[\beta_{f_t,t}].$$

**Bounding the Confidence Intervals**     The packet selected by $\mathrm{ALG}^{Rs}$ is random, thus, we rewrite

$$\sum_{t=1}^{T} \mathbb{E}_{x_t}[\beta_{f_t,t}] = \sum_{t=1}^{T} \sum_{i \in [K]} \mathbb{P}_t(f_t = i)\beta_{i,t_{i,j}},$$

where $\mathbb{P}_t(f_t = i)$ is the probability that the selected packet is $i$ at time $t$. Let $N_{i,t}$ be the number of times a packet of type $i$ has been selected up to time $t$. We consider a worst-case allocation, where we maximize the probability of the packet with the largest UCB to be selected. Then

$$\sum_{t=1}^{T} \sum_{i \in [K]} \mathbb{P}_t(f_t = i)\beta_{i,t} \leq \sum_{t=1}^{T} \max_{i \in [K]} \beta_{i,t}$$

$$= \sum_{t=1}^{T} c \max_{i \in [K]} \sqrt{\frac{\ln \frac{KT}{\delta}}{N_{i,t}}}$$

$$\leq \sum_{t=1}^{T} c \sqrt{\frac{\ln \frac{KT}{\delta}}{\frac{t}{K}}}$$

$$\leq \widetilde{\mathcal{O}}\left( \sqrt{KT} \right),$$

where we used the fact that the worst-case allocation is performing round-robin, where the $\widetilde{\mathcal{O}}$ only hides universal constants and logarithmic terms.

Setting $\delta = \frac{1}{T}$, the good event holds with probability at least $1 - \frac{1}{T}$:

$$\mathbb{E}[R_{\frac{e}{e-1},T}^{\mathrm{ALG}^{Rs}}] = \mathbb{E}[R_{\frac{e}{e-1},T}^{\mathrm{ALG}^{Rs}}|\mathcal{E}(\delta)]\mathbb{P}\left(\mathcal{E}(\delta)\right) + \mathbb{E}[R_{\frac{e}{e-1},T}^{\mathrm{ALG}^{Rs}}|\mathcal{E}(\delta)^C]\mathbb{P}\left(\mathcal{E}(\delta)^C\right)$$
$$\leq \widetilde{\mathcal{O}}\left(\sqrt{KT}\right).$$

Putting all together completes the proof. $\qquad\square$

