# OpenReview forum: "Online Packet Scheduling with Deadlines and Learning"
_ICML.cc/2026/Conference — ICML 2026 regular_

### Official Review · Reviewer_33WS · 2026-02-17

**Soundness:** 3
**Presentation:** 2
**Significance:** 3
**Originality:** 3
**Overall Recommendation:** 4
**Confidence:** 3

**Summary:**

The paper studies online packet scheduling problems with deadlines.
In the classical problem, packets (jobs) with unit processing time arrive online at integer times and have a weighted and a deadline. During each timestep, one packet can be scheduled. A packet can only be scheduled before its deadline. The goal is to maximize the total weight of scheduled packets.

In terms of competitive ratio, a Golden-Ratio-competitive deterministic algorithm and stronger randomized algorithms are known.
This paper considers a variant with less information, namely that the weights are unknown until the package is scheduled. However, there is a weight distribution for packet types known, and each packet belong to one of those K types. Hence, the problem now becomes an online learning problem where the goal is to learn those weight distributions and at the same time make good decisions in scheduling packets.

The authors present different results in terms of $\alpha$-regret. All learning bounds dependent on $\sqrt{T}$, where T is the time horizon, which is best-possible, as the problem captures variants of the classical MAB problem, in particular the sleeping bandit problem.
Moreover, the authors present improved results in the regime where the deadline is close to the arrival of a packet. This scenario is also well motivated in the paper.

Combining online learning with classical online algorithms has become popular in recent years, and similar examples of such results for preemptive scheduling or caching have been studied before. Hence, the problem and results should be of interest to the ICML community.

**Compliance With Llm Reviewing Policy:**

Affirmed.

**Final Justification:**

I keep my initial assessment and think that the paper should have a chance of being accepted.

**Key Questions For Authors:**

- Can you give more details on related work?
- How does this problem related to general weighted throughput maximization?
- How do the techniques compare to other problems where online learning and scheduling have been combined?

**Limitations:**

yes

**Strengths And Weaknesses:**

### Strengths:

- Introduction of a new model for connecting online learning and packet scheduling, which is well motivated.
- Strong learning bounds in all regimes.
- Improved competitive ratios for x-bounded regimes, which should be of independent interest for the online scheduling community.
- Good fit for ICML, interesting for multiple sub-communities.


### Weaknesses:

- As many online learning results, the results are based on standard UCB-type algorithms. However, this feels unavoidable if one wants to achieve tight guarantees.
- There should be more discussion about related work: online learning, scheduling, learning-augmented algorithms, other examples of where online learning and online scheduling have been combined, with more details.


### Minor comments:
- You should explain better what 2-bounded and 3-bounded means (3-bounded is not defined before section 2). Generally, I felt the intro a bit hard to read, as there are many different properties and parameters one needs to be aware of. I would appreciate if this could be improved, but I also have no concrete suggestions.
- L64: missing whitespace: "algorithms.In"
- L81: alpha-no-regret?

---

> ### Author Rebuttal · Authors · 2026-03-31
>
> We thank the Reviewer for the time and consideration in reviewing our work. Also, we thank the Reviewer for the positive evaluation of our work and the kind words on the strength of our results and on the paper being a good fit within ICML.
>
> > **The results are based on standard UCB-type algorithms**
>
> Our randomized algorithms make use of both UCBs and lower confidence bounds (LCBs). Indeed, LCBs enable randomized algorithms to handle ratios of estimates more naturally. This emerges from the proofs of Theorems 5.1 and 5.2.
>
> This discussion is reported in Section 6 of the paper, as well as some additional technical challenges that greatly differentiate our analyses from the standard bandit literature. In particular:
>
> * Buffer Divergence between ALG and OPT: in MABs, an algorithm can always choose the same action as OPT. In K-OPSD, a single different decision leads to different buffers in the subsequent rounds. Thus, it is not possible, in general, to bound the instantaneous regret.
> * Breaking the $\Phi$ barrier for the competitive ratio in 2-bounded instances: this contribution is not directly related to the learning problem, but rather to the standard OPSD without learning, and we believe it is of great interest in its own, as it was proven using unseen techniques in the literature.
> * Providing a $\theta_{K-1}$-regret lower bound: proving a lower bound on the regret in bandits usually requires two instances that are hard to distinguish, but have different enough optimal strategies. On the other hand, proving a lower bound on the competitive ratio for OPSD requires a carefully crafted instance where every decision always leads to the same competitive ratio. The two objectives collide: two instances where every decision always leads to the same outcome cannot be different enough to generate a meaningful regret lower bound. We overcome this problem by proposing a particular three-instance construction.
>
> > **Related Works and Relationship with General Weighted Throughput Maximization**
>
> In the broader scheduling literature, the Online Packet Scheduling with Deadlines (OPSD) problem is mathematically equivalent to maximizing the weighted throughput of unit-size jobs. In the table that follows, we report the existing results on the general problem where packets might have a non-unitary size or where packets have unit weights. Results for OPSD can be added to this table as "Weighted, Unit Sizes".
>
> | Setting | Algorithm Type | Competitive Ratio / Performance Bound
> | :--- | :--- | :--- |
> | **Unit Weights, Arbitrary Sizes** | Det. | $\Theta(2)$ [1] |
> | **Weighted, Bounded Sizes** ($< k$) | Det. | $\Theta\left(\frac{k}{\log k}\right)$ [2] |
> | **Weighted, Exact Sizes**( $= k$) | Det.  | $\Omega(2.598)$, $\mathcal{O}\left(5\right)$ [2] |
> | **Unit Weights, Bounded Sizes** ($\le k$, $w_j = 1 ~~\forall j$) | Det.  | $\Omega(\ln K/ \ln \ln K)$ [2], $\mathcal{O}\left(\ln K\right)$ [3] |
> [1] Hoogeveen, H., Potts, C. N., & Woeginger, G. J. (2000). On-line scheduling on a single machine: maximizing the number of early jobs. Operations Research Letters, 27(5), 193-197.
>
> [2] Dürr, C., Jeż, Ł., & Nguyen, K. T. (2009). Online scheduling of bounded length jobs to maximize throughput. In International Workshop on Approximation and Online Algorithms (pp. 116-127).
>
> [3] Kalyanasundaram, B., & Pruhs, K. (1998, August). Maximizing job completions online. In European Symposium on Algorithms (pp. 235-246). Berlin, Heidelberg: Springer Berlin Heidelberg.
>
> > **How do the techniques compare to other problems where online learning and scheduling have been combined?**
>
> The closest prior works are technically distinct. This expanded discussion is already cited and will be included in the final version:
>
> * **Merlis et al. (2023):** Relies on *preemption* in stochastic scheduling. In contrast, our setting features *unit-size packets, hard deadlines, adversarial arrivals, and post-service bandit feedback*. Because early mistakes alter future buffers, we replace standard regret decompositions with tailored *competitive-analysis charging arguments* and confidence bounds.
> * **Levy et al. (2024):** Learns unknown costs via *fractional paging solvers and randomized rounding*. Our randomized algorithms must instead control *ratios of estimates* within deadline-based rules, necessitating both UCBs and LCBs.
> * **Liang et al. (2023) & Karavasilis (2025):** Both leverage exogenous predictions to bypass classical lower bounds (e.g., Karavasilis achieves bounds of $OPT-\eta$ in online interval scheduling using binary predictions). Our setting assumes *no exogenous predictions*; everything is learned from realized feedback.
>
> Ultimately, our contribution is not merely adding learning to scheduling, but developing a new synthesis of *bandit estimation + competitive packet-scheduling analysis* for deadline-constrained buffers.
>
> [4] Karavasilis, C. (2025). Interval selection with binary predictions. arXiv preprint arXiv:2502.10314.

---

> > ### Author Rebuttal · Reviewer_33WS · 2026-03-31
> >
> > I thank the authors for their detailed rebuttal and for answering my questions.
> >
> > I will keep my opinion that this paper should have a chance of being accepted.

---

### Official Review · Reviewer_Gn4m · 2026-03-09

**Soundness:** 3
**Presentation:** 3
**Significance:** 2
**Originality:** 2
**Overall Recommendation:** 4
**Confidence:** 3

**Summary:**

This paper studies an online scheduling problem with deadlines under partial feedback. In each time slot, the algorithm can schedule at most one packet. Each packet has an arrival time, a deadline, and a type. The reward of a packet is drawn from an unknown distribution associated with its type, and is only observed if that packet is scheduled. The paper formalizes this setting as K-OPSD and uses α-regret as the main evaluation metric, with the goal of capturing both the unavoidable loss caused by deadline-constrained online scheduling and the additional loss caused by learning.

The paper first shows that the 1-bounded case is equivalent to sleeping bandits. It then provides several theoretical results. For deterministic 2-bounded and 3-bounded settings, it gives instance-dependent and worst-case Φ-regret bounds. For the 2-bounded setting with finitely many types, it proposes a dynamic-threshold algorithm that improves the classical deterministic competitive ratio from Φ to θ_K, and further derives a learning version with θ_K-regret upper bounds and corresponding lower bounds. For randomized settings, it gives a 5/4-regret algorithm for 2-bounded instances and an e/(e−1)-regret algorithm for general s-bounded instances, both with an additional $\widetilde O(\sqrt{KT})$ term. Overall, this is a theory-driven paper focused on modeling, algorithm design, and analysis, without empirical evaluation.

**Compliance With Llm Reviewing Policy:**

Affirmed.

**Final Justification:**

The paper studies a clear and technically solid problem at the intersection of online packet scheduling and bandit learning. The formulation is well defined, and the main theoretical results appear sound and meaningful for this structured setting.

My main concern was the strength and practical meaning of the fixed K-type assumption. The rebuttal addressed this concern adequately by clarifying both its tractability role and its connection to class-based traffic modeling in QoS systems. While the final version could still clarify the scope of the abstraction more explicitly, my main concern has been resolved. Taking both the paper and the rebuttal into account, I am updating my recommendation to 4.

**Key Questions For Authors:**

- How realistic is the fixed K-type assumption in the intended application domains? In particular, is it reasonable to assume that all packets come from a fixed set of K categories with shared reward distributions?
- I understand why the paper introduces K types: without such sharing, the problem would not be learnable in the usual bandit sense, since each packet can only be scheduled once. However, this also suggests that the learnability of the model depends heavily on this assumption. Could the authors clarify whether they view this as a natural modeling assumption or primarily as a tractability assumption?
- Would a contextual or linear bandit formulation be more appropriate for this setting? For example, one could imagine that each packet has its own context, while rewards are generated through a shared unknown parameter. Such a model would preserve packet-level heterogeneity while still allowing statistical sharing across packets. Do the authors believe such a formulation would better capture the intended applications?
- More broadly, what aspects of the intended packet scheduling applications are preserved by the K-type abstraction, and what important aspects are abstracted away?
- The paper motivates the model using QoS and advertising examples, but does not provide empirical or domain-specific evidence supporting the K-type abstraction. Can the authors provide more justification that this modeling choice is meaningful in practice?

**Limitations:**

- The main limitation of the paper lies in the strength of its modeling assumptions. In particular, the learning formulation depends critically on the assumption that all packets belong to a fixed set of K known types, with packets of the same type sharing a common reward distribution. This assumption makes the problem learnable, but also narrows the scope of the model and moves it closer to a dynamic-availability bandit setting. As a result, the theoretical results are best understood as applying to a strongly structured abstraction of packet scheduling with unknown rewards, rather than to a fully general version of the problem. The paper would benefit from a clearer discussion of this tradeoff, as well as a more explicit account of the practical settings in which the K-type abstraction is appropriate.

**Strengths And Weaknesses:**

### Strengths

- The paper is well motivated, and its main claims are broadly aligned with its formal model and theoretical analysis. The definition of K-OPSD is clear, and the introduction of α-regret has a coherent technical motivation. The theoretical results are organized around a well-defined problem setting, and the relationship between the proposed algorithms, the benchmark, and the guarantees is mostly consistent.

### Weaknesses

- My main concern is that the problem setting relies on a rather strong structural assumption in order to make learning possible. In a standard bandit problem, one learns an arm by pulling it multiple times. In this packet scheduling setting, however, an individual packet can only be scheduled once, so there is no natural repeated sampling at the packet level. To address this, the paper assumes that all packets come from a fixed set of K types, and that packets of the same type share the same reward distribution. This makes learning feasible, but it also substantially changes the nature of the original problem: the uncertainty is no longer packet-specific, but type-specific.

- Because of this modeling choice, the problem becomes much closer to a K-armed bandit with dynamic availability, especially in light of the connection to sleeping bandits. At a high level, one is learning K shared reward distributions, while the available instances appear and disappear over time. This does not mean the scheduling structure disappears, but it does make the learning component look much more like a dynamic-arm-set bandit problem than a fully general packet scheduling problem with unknown values. As a result, I am not fully convinced that the paper preserves the distinctive difficulty of packet scheduling to the extent suggested by its framing.

- Relatedly, the K-type assumption appears quite strong from an application perspective. The paper mentions QoS networks and digital advertising as motivating examples, but it does not provide evidence that such applications naturally satisfy a fixed, known, small-K type structure with shared stationary reward distributions. Without such evidence, it is difficult to judge whether this abstraction is a realistic modeling step or mainly a technical device introduced to make the learning problem

---

> ### Author Rebuttal · Authors · 2026-03-30
>
> We thank the Reviewer for their time, constructive feedback, and kind words on the setting's relevance and formalization. We will include these discussions in the final paper.
>
> > **Q1. How realistic is the fixed $K$-type assumption in the intended application domains? / Q5. more justification that this modeling choice is meaningful in practice?**
>
> The $K$-type assumption is a necessary tractability requirement and a highly realistic model of standard networking practices. In our intended domains, traffic is rarely treated as purely heterogeneous per packet. To our understanding, it is aggregated into discrete, manageable classes with shared statistical properties, mapping directly to modern QoS architectures:
>
> - **Differentiated Services (DiffServ) and Traffic Classification**: Tracking individual flows is unscalable, so IP networks rely on DiffServ [1,2] for QoS. DiffServ classifies packets into a limited number of discrete traffic classes [3]. Routers manage traffic strictly based on this finite set, which is the exact realization of our $K$-type assumption.
> - **Protocol-Level Support (IPv6 Traffic Class)**: This classification is hardcoded into internet protocols. The IPv6 Traffic Class field [4] uses 6 bits for the Differentiated Services Code Point (DSCP), providing up to 64 codepoints ($K \le 64$). Administrators use these $K$ values to mark packets with standard priority codes (e.g., Expedited Forwarding for VoIP, Assured Forwarding for video).
> - **Kernel-Level Schedulers (Linux Traffic Control)**: At the OS level, traffic control is managed via Queuing Disciplines (qdisc). Classful qdiscs (e.g., Hierarchical Token Bucket) use filters to classify packets into a predefined, finite set of classes [5] based on the DSCP field, where the scheduler applies specific priority and bandwidth rules.
>
> Clustering packets with similar features makes traffic-intensive routers manageable. We believe the $K$-type model is thus more common in practice than standard OPSD.
>
> [1] https://en.wikipedia.org/wiki/Differentiated_services
> [2] Carpenter, B. E., & Nichols, K. (2002). Differentiated services in the Internet. Proceedings of the IEEE, 90(9), 1479-1494.
> [3] https://en.wikipedia.org/wiki/Traffic_classification
> [4] https://en.wikipedia.org/wiki/IPv6_packet#Fixed_header
> [5] https://docs.redhat.com/en/documentation/red_hat_enterprise_linux/8/html/configuring_and_managing_networking/linux-traffic-control_configuring-and-managing-networking
>
> > **Q3. Would a contextual or linear bandit formulation be more appropriate for this setting?**
>
> We agree this is a very promising direction, and we mention it as future work. Such models would preserve packet-level heterogeneity while enabling statistical sharing across packets. However, we view them as a strict generalization of our setting, not a replacement. In particular, we believe that understanding our setting is a necessary first step before tackling contextual/linear versions: even in this simpler setting, one must already resolve the interaction between deadline-constrained scheduling, partial feedback, and learning. Our goal here is precisely to isolate and solve this core difficulty first, and then build on it toward richer contextual models.
>
> > **Q2. Natural modelling assumption or primarily as a tractability assumption? / Q4. More broadly, what aspects of the intended packet scheduling applications are preserved by the K-type abstraction, and what important aspects are abstracted away?**
>
> It is fundamentally both. $K$-OPSD precludes observing a packet's weight before scheduling. Without the $K$-type assumption (i.e., $K=\infty$), the problem is unmanageable: regret becomes linear and the competitive ratio grows arbitrarily (no policy beats a random one). The $K$-type assumption bridges this gap, moving away from worst-case guarantees of standard OPSD (which assumes fully observable weights) while maintaining mathematical tractability for sub-linear regret.
>
> This makes the problem **closer** to real-world applications where packets cluster into finite types, but exact weights are unobservable beforehand.
>
> The $K$-type abstraction preserves key challenges: online arrivals, hard deadlines, partial feedback, adversarial sequences, and scheduling loss (competitive ratio). It abstracts away within-type heterogeneity and nonstationarity.

---

> > ### Author Rebuttal · Reviewer_Gn4m · 2026-04-06
> >
> > Thank you for the detailed rebuttal. My concerns have been adequately addressed, and I have decided to increase my score to 4.

---

### Official Review · Reviewer_vf3W · 2026-03-11

**Soundness:** 4
**Presentation:** 3
**Significance:** 3
**Originality:** 3
**Overall Recommendation:** 4
**Confidence:** 3

**Summary:**

This paper studies online packet scheduling with deadlines when packet rewards are unknown and only revealed upon transmission. It combines competitive analysis with bandit learning, and shows that sublinear $\alpha$-regret is achievable in several settings while matching classical competitive ratios. For deterministic 2-bounded instances with finitely many packet classes, the paper improves the classical golden-ratio barrier to a tighter ratio.

**Compliance With Llm Reviewing Policy:**

Affirmed.

**Final Justification:**

I thank the authors for their detailed response. I decide to maintain my positive score.

**Key Questions For Authors:**

The result improving the classical deterministic ratio from $\Phi$ to $\theta_K < \Phi$ is very interesting. Is there an intuitive structural explanation for why finitely many packet types make this improvement possible, beyond the technical threshold construction?

**Limitations:**

yes

**Strengths And Weaknesses:**

Strength:
1. The problem formulation is novel and well-motivated. The paper introduces K-OPSD, combining deadline-constrained online packet scheduling with bandit-style partial feedback.
2. The paper gives sublinear $\alpha$-regret guarantees in several important regimes and claims these match the standard bandit lower-bound order.

Weakness:
1. Rewards are class-dependent with one unknown distribution per packet type, which is theoretically well-defined but may not capture realistic contextual structure.
2. Many of the strongest results rely on special cases such as 2-bounded or 3-bounded instances, and tight $\Phi$-regret for general deterministic instances remains open.

---

> ### Author Rebuttal · Authors · 2026-03-30
>
> We thank the Reviewer for the time and consideration in reviewing our work. Also, we thank the Reviewer for the positive evaluation of our work and the kind words on the setting's novelty and relevance.
>
> In what follows, we address point-by-point the Reviewer's concerns.
>
> **The setting may not capture realistic contextual structure**
> Our work has the goal to be the first to explore a nice and practically relevant generalization of both the OPSD problem and the Sleeping Bandit problem, by showing how techniques from the two worlds can be combined to obtain meaningful theoretical guarantees. As a future research direction, it would be interesting to investigate settings such as the linear $K$-OPSD setting and the contextual $K$-OPSD setting. In particular, we believe that understanding our setting is a necessary first step before tackling contextual/linear versions. We will include this discussion in the final version of the paper.
>
> **Tight $\Phi$-regret for general deterministic instances remains open.**
> The only unified deterministic algorithm for OPSD is the one from [Veselỳ et al., 2022], achieving the tight competitive ratio of $\Phi$ for every degree of slackness. Finding such an algorithm has been an open problem for more than 15 years, as it requires an extraordinarily complex combination of techniques. On the other hand, specific (and mostly studied) cases such as 2 and 3-bounded instances can be dealt with using simpler EDF-style algorithms. We consider beyond the scope of this work to adapt the algorithm from [Veselỳ et al., 2022] to the learning case, as it would require a huge technical effort to reproduce that algorithm while dealing with uncertainty in the estimation,  without a significant additional methodological novelty. As stated in the previous answer, our goal is to be the first to explore a nice intersection between two seemingly unrelated research areas, which we consider an important contribution. Of course, an interesting follow-up to our paper would be to provide a deterministic no-$\Phi$-regret algorithm for every degree of slackness. We will include this discussion in the final version of the paper.
>
> **The result improving the classical deterministic ratio from $\Phi$ to $\theta_K<\Phi$ is very interesting. Is there an intuitive structural explanation for why finitely many packet types make this improvement possible, beyond the technical threshold construction?**
>
> The best way to get an intuition on this is to explore the lower bound construction from [Hajek]. To enforce a competitive ratio of $\Phi$, the adversary has to rely on the _infinite trick_, i.e. a potentially infinite sequence of packets with different weights that can put every algorithm in a sequence of infinite trade-offs. Every time $t$, the algorithm is faced with a lighter packet with deadline $t$ and a heavier one with deadline $t+1$. Choosing the former means going on with the sequence, where the adversary generates a new packet with deadline $t+2$ heavier than the other. Asymptotically, if the algorithm continues to choose the one with the earliest deadline, the competitive ratio converge to $\Phi$. Instead, choosing the latter means stopping the sequence, but the weights have been carefully crafted to force a competitive ratio of $\Phi$ at any point of it. Having a finite number of weights precludes the adversary to go on with this trick infinite times, as the new packet arriving in every round must have a greater weight than the pre-existing ones to present a trade-off. Moreover, the adversary is now forced to modulate the $K$ available weights to force the worst-possible competitive ratio of $\theta_K$ in case the algorithm chooses the packet with deadline $t+1$.
>
> From an algorithmic perspective, it is interesting to see that a static $EDF_{\theta_K}$ algorithm is not capable of matching the competitive ratio of $\theta_K$, and it can be seen by using the [Hajek] instance. Our algorithm combines epoch-based dynamic thresholding with a novel analysis to achieve the tight competitive ratio $\theta_K$.
>
> This discussion will be expanded in the final version of the paper.

---

> > ### Author Rebuttal · Reviewer_vf3W · 2026-04-03
> >
> > I thank the authors for their detailed response. I decide to maintain my positive score.

---

### Official Review · Reviewer_Xesy · 2026-03-12

**Soundness:** 3
**Presentation:** 4
**Significance:** 4
**Originality:** 3
**Overall Recommendation:** 5
**Confidence:** 3

**Summary:**

This paper considers a learning augmented model for a classic online problem: packet scheduling with deadlines (OPSD). In the original problem, packets arrive over time with each packet having a weight and deadline. One needs to send <= 1 packet per time step so as to maximize the weight of packets sent before their deadline. In this paper, they consider a setting with K weight classes, where all packets from a particular class i have weight drawn from an unknown distribution D_i. They obtain A-approximate regret bounds which bound the additive error between the algorithm’s weight with A times the optimum; here A is the best competitive ratio known for usual OPSD.

**Compliance With Llm Reviewing Policy:**

Affirmed.

**Key Questions For Authors:**

1) Can you obtain a unified result that covers all the cases (at least for deterministic algorithms)?
2) Isn't a 2-regret of \sqrt{T} for the unbounded-slack setting immediate from the OPSD algorithm from [Hajek]? Your table has NA here.

**Limitations:**

yes

**Strengths And Weaknesses:**

The authors consider a variety of settings: s-bounded instances (where deadline is always within s steps) and deterministic/randomized algorithms. For each setting they use the best known online algorithm (in the full info case) and adapt it by using UCB-estimates for the weights. They also provide matching lower bounds for some cases, which is interesting because such results are rare for approximate regret bounds. Their formulation also leads to the question whether better competitive ratios are possible for OPSD instances with a small number K of types: they show that this is indeed the case for 2-bounded instances, and also extend it to their learning setting.

I found the results interesting. Although the UCB-approach is the natural one, the analysis needs to combine these ideas carefully with the OPSD analysis.

One unsatisfactory issue is that they need a separate analysis for each of the learning-augmented settings. It would be much better if they could abstract a property of the OPSD online algorithm under which their UCB-based method guarantees an A-regret bound of \sqrt{T}. Some such results have appeared recently for stochastic optimization, e.g.
1) Arpit Agarwal, Rohan Ghuge, Viswanath Nagarajan: Semi-Bandit Learning for Monotone Stochastic Optimization. FOCS 2024: 1260-1274
2) Rad Niazadeh, Negin Golrezaei, Joshua R. Wang, Fransisca Susan, Ashwinkumar Badanidiyuru: Online Learning via Offline Greedy Algorithms: Applications in Market Design and Optimization. Manag. Sci. 69(7): 3797-3817 (2023)

---

> ### Author Rebuttal · Authors · 2026-03-30
>
> We thank the Reviewer for the time and consideration in reviewing our work. Also, we thank the Reviewer for the positive evaluation of our work.
>
> In what follows, we address point-by-point the Reviewer's concerns.
>
> **Can you obtain a unified result that covers all the cases (at least for deterministic algorithms)?**
>
> The only unified deterministic algorithm for OPSD is the one from [Veselỳ et al., 2022], achieving the tight competitive ratio of $\Phi$ for every degree of slackness. Finding such an algorithm has been an open problem for more than 15 years, as it requires an extraordinarily complex combination of techniques. On the other hand, specific (and mostly studied) cases such as 2 and 3-bounded instances can be dealt with using simpler EDF-style algorithms. We consider beyond the scope of this work to adapt the algorithm from [Veselỳ et al., 2022], to the learning case, as it would require a huge technical effort to reproduce that algorithm while dealing with uncertainty in the estimation, without a significant additional methodological novelty. Our work has the goal to be the first to explore a nice and practically relevant generalization of both the OPSD problem and the Sleeping Bandit problem, by showing how techniques from the two worlds can be combined to obtain meaningful theoretical guarantees. Of course, an interesting follow-up to our paper would be to provide a deterministic no-$\Phi$-regret algorithm for every degree of slackness. We will include this discussion in the final version of the paper.
>
> **Isn't a 2-regret of \sqrt{T} for the unbounded-slack setting immediate from the OPSD algorithm from [Hajek]? Your table has NA here.**
>
> Yes, it is possible to get a 2-regret of $\sqrt{T}$ by using the AUER algorithm from [Kleinberg et al., 2010]. We have formalized the argument (which combines the argument from [Hajek] with standard optimism) and we will include that in the paper. We thank the Reviewer for having spotted this point, as it makes a nice addition to the results table.

---

> > ### Author Rebuttal · Reviewer_Xesy · 2026-04-03
> >
> > Thanks for the response. It validates my original rating.

---

### Decision · Program_Chairs · 2026-04-30

**Decision:**

Accept (regular)

**Comment:**

The paper was overall received positively by the reviewers. There was a concern about practical relevance of the K-type assumption, but it was clarified in the rebuttal. The authors are also encouraged to expand the discussion of related work, as provided in their rebuttal. I recommend acceptance.